 eLIFE

# *Usf1*, a suppressor of the circadian *Clock* mutant, reveals the nature of the DNA-binding of the CLOCK:BMAL1 complex in mice

Kazuhiro Shimomura[1], Vivek Kumar[2], Nobuya Koike[3], Tae-Kyung Kim[3], Jason Chong[4], Ethan D Buhr[4], Andrew R Whiteley[4], Sharon S Low[4†a], Chiaki Omura[5], Deborah Fenner[5], Joseph R Owens[6], Marc Richards[4], Seung-Hee Yoo[2], Hee-Kyung Hong[5], Martha H Vitaterna[1], Joseph Bass[7], Mathew T Pletcher[8†b], Tim Wiltshire[8†c], John Hogenesch[9], Phillip L Lowrey[4†d], Joseph S Takahashi[2]*

[1]Center for Functional Genomics, Department of Neurobiology, Center for Sleep and Circadian Biology, Northwestern University, Evanston, United States; [2]Department of Neuroscience, Howard Hughes Medical Institute, University of Texas Southwestern Medical Center, Dallas, United States; [3]Department of Neuroscience, University of Texas Southwestern Medical Center, Dallas, United States; [4]Department of Neurobiology, Northwestern University, Evanston, United States; [5]Department of Neurobiology, Center for Functional Genomics, Northwestern University, Evanston, United States; [6]Department of Neurobiology, Center for Sleep and Circadian Biology, Northwestern University, Evanston, United States; [7]Department of Medicine, Northwestern University Feinberg School of Medicine, Chicago, United States; [8]Department of Genomics, Genomics Institute of the Novartis Research Foundation, San Diego, United States; [9]Department of Pharmacology, Perelman School of Medicine, University of Pennsylvania, Philadelphia, United States

*For correspondence: joseph.takahashi@utsouthwestern.edu

†Present address: [a]Center for Scientific Review, National Institutes of Health, Bethesda, United States; [b]Rare Diseases Research Unit, Pfizer Worldwide Research and Development, Cambridge, United States; [c]Division of of Pharmacotherapy and Experimental Therapeutics, UNC Eshelman School of Pharmacy, Chapel Hill, United States; [d]Department of Biology, Rider University, Lawrenceville, United States

**Competing interests:** The authors declare that no competing interests exist.

**Abstract** Genetic and molecular approaches have been critical for elucidating the mechanism of the mammalian circadian clock. Here, we demonstrate that the *ClockΔ19* mutant behavioral phenotype is significantly modified by mouse strain genetic background. We map a suppressor of the *ClockΔ19* mutation to a ~900 kb interval on mouse chromosome 1 and identify the transcription factor, *Usf1*, as the responsible gene. A SNP in the promoter of *Usf1* causes elevation of its transcript and protein in strains that suppress the *Clock* mutant phenotype. USF1 competes with the CLOCK:BMAL1 complex for binding to E-box sites in target genes. Saturation binding experiments demonstrate reduced affinity of the CLOCKΔ19:BMAL1 complex for E-box sites, thereby permitting increased USF1 occupancy on a genome-wide basis. We propose that USF1 is an important modulator of molecular and behavioral circadian rhythms in mammals.

## Introduction

To adapt to daily environmental cycles, most organisms have evolved endogenous clocks composed of cell-autonomous, self-sustained oscillators that drive 24-hr rhythms in biochemistry, physiology and behavior (*Bass and Takahashi, 2010*; *Lowrey and Takahashi, 2011*). In mammals, the innate periodicity of the circadian clock is generated by transcriptional/translational feedback loops composed of a core set of clock genes expressed in cells throughout the body (*Reppert and Weaver, 2002*; *Lowrey and Takahashi, 2011*). Two members of the bHLH-PAS transcription factor family, CLOCK (and its

---

**eLife digest** Circadian rhythms are biochemical, physiological and behavioral processes that follow a 24-hr cycle, responding primarily to the periods of light and dark, and they have been observed in bacteria, fungi, plants and animals. The circadian clock that drives these rhythms—which dictate our sleep patterns and other processes—involves a set of genes and proteins that participate in a collection of positive and negative feedback loops.

Previous research has mainly focused on identifying core clock genes—that is, genes that make up the molecular clock—and studying the functions of these genes and the proteins they code for. However, it has become clear that other clock genes are also involved in circadian behavior, and it has been proposed that polymorphisms in these non-core clock genes could contribute to the variations in circadian behavior displayed by different mammals.

One important feedback loop in mammals involves two key transcription factors, CLOCK and BMAL1, that combine to form a complex that initiates the transcription of the negative feedback genes, *Period* and *Cryptochrome*. Shimomura et al. discovered that *Usf1*, a gene that codes for a transcription factor that is typically involved in lipid and carbohydrate metabolism, as well as other cellular processes, is also important. In particular, this transcription factor is capable of partially rescuing an abnormal circadian rhythm caused by a mutation in the *Clock* gene in mice.

Shimomura et al. showed that the proteins expressed by the mutant *Clock* gene can bind to the same regulatory sites in the genome as the normal CLOCK:BMAL1 complex, but that gene expression of these targets is reduced because transcriptional activation is lower and binding of the complex is not as strong. However, proteins expressed by the *Usf1* gene are able to counter this by binding to the same sites in the genome and compensating for the mutant CLOCK protein. Further experiments are needed to explore how the interactions between the USF1 and CLOCK:BMAL1 transcriptional networks regulate circadian rhythms and, possibly, carbohydrate and lipid metabolism as well.

---

paralog NPAS2) and BMAL1 (ARNTL), heterodimerize and initiate transcription of the *Period* (*Per1, Per2*) and *Cryptochrome* (*Cry1, Cry2*) genes through E-box regulatory sequences (***King et al., 1997***; ***Gekakis et al., 1998***; ***Hogenesch et al., 1998***; ***Kume et al., 1999***; ***Bunger et al., 2000***; ***Huang et al., 2012***). The resulting PER and CRY proteins form multimeric complexes, translocate to the nucleus, and abrogate their own transcription by repressing CLOCK:BMAL1 (***Lee et al., 2001***; ***Koike et al., 2012***). As the PER and CRY proteins are degraded, CLOCK:BMAL1 occupancy increases to initiate a new round of transcription (***Lowrey and Takahashi, 2011***; ***Koike et al., 2012***). While this forms the major feedback loop of the mammalian molecular clock, the overall mechanism also depends on additional feedback loops driven by ROR and REV-ERBα/β (***Preitner et al., 2002***; ***Sato et al., 2004***; ***Ueda et al., 2005***; ***Cho et al., 2012***).

Significant progress has been made in identifying the core clock genes and the functions of their protein products in the clock mechanism, yet it is clear from other studies, including mutagenesis screens, that additional genes are necessary for a fully functional circadian clock (***Takahashi, 2004***). We previously identified at least 13 loci in mice that affect circadian behavior through complex epistatic interactions (***Shimomura et al., 2001***), indicating that there are other clock-relevant genes in the mammalian genome. By extension, it is possible that the variance in circadian behavior observed in the human population results from polymorphisms in non-core circadian clock genes. Thus, identifying these genes is important for both mechanistic understanding and translational application.

The quantitative trait locus (QTL) approach has been successfully applied to detect loci that associate with phenotypes of interest. An important application of QTL analysis is the identification of loci that modify the function and/or expression of a particular gene (***Nadeau and Topol, 2006***). A major obstacle, however, has been the cloning of these genes—particularly those underlying behavioral QTLs (***Nadeau and Frankel, 2000***). To overcome this challenge, several approaches have been proposed, yet it remains difficult to obtain a mapping resolution suitable for gene cloning by the positional candidate method (***Darvasi and Soller, 1995***; ***Churchill et al., 2004***; ***Valdar et al., 2006***). To date there are few examples of behavioral QTLs having been cloned using high-resolution mapping strategies (***Yalcin et al., 2004***; ***Watanabe et al., 2007***; ***Tomida et al., 2009***).

## Results

The mouse *Clock^Δ19* mutation was originally induced on the C57BL/6J (hereafter B6) isogenic background by ENU mutagenesis (*Vitaterna et al., 1994*). *Clock^Δ19* lengthens circadian free-running period by about 1 hr in heterozygous mice, and by about 4 hr in homozygous mutant mice followed by arrhythmicity upon exposure to constant darkness (DD). We observed that the circadian period phenotype in *Clock^Δ19*/+ animals is suppressed by crossing to the BALB/cJ (hereafter BALB) inbred strain. On an isogenic B6 background, *Clock^Δ19*/+ lengthens period ~0.6 hr, and this is suppressed to ~0.3 hr on a (BALB x B6)F1 background. In ([BALB x B6]F1 x BALB)N4 mice, the *Clock^Δ19*/+ period phenotype is completely suppressed (*Figure 1A,B*). This suggests that at least one dominant suppressor allele exists in the BALB genome. To test whether this suppression is also cell or tissue autonomous, we monitored the circadian period of bioluminescence in *Clock^Δ19*/+ suprachiasmatic nucleus (SCN) and pituitary explants from *Period2^Luciferase* (*Per2^Luc*) reporter mice (*Yoo et al., 2004*). The circadian period in both *Clock^Δ19*/+ SCN and pituitary was shorter in (BALB x B6)F1 animals than in mice of the B6 background (*Figure 1C*). These results demonstrate that the BALB suppression phenotype is tissue autonomous and present in both central (SCN) and peripheral (pituitary) circadian clocks.

To assess whether suppressor alleles segregate in the BALB x B6 hybrids, we analyzed the circadian locomotor activity rhythm of *Clock^Δ19*/+ mice from five different populations between the BALB and B6 strains (*Figure 1D*). Analysis of period in these crosses suggests that the BALB allele acts semidominantly to suppress the *Clock^Δ19* mutation. To map the *Clock* suppressor loci, we used the (BALB x B6) F2 generation (*Figure 1D*). We performed QTL analysis on 222 (BALB x B6)F2 *Clock^Δ19*/+ mice with 87 simple sequence length polymorphism markers (SSLP) and detected a significant association between *Clock^Δ19* phenotype suppression and a locus on mouse chromosome 1, which we named *Suppressor of Clock* (*Soc*) (*Figure 1E*).

To isolate this region, we first selected mice carrying large fragments of BALB chromosome 1 by genotyping five SSLPs spanning this region (*D1Mit22*, 59.95 Mb–*D1Mit155*, 196.25 Mb). The congenic region was progressively narrowed to 25 Mb flanked by *D1Mit452* (165.10 Mb) and *D1Mit223* (190.45 Mb) (*Figure 1E*). We observed a significantly shorter free-running period in *Clock^Δ19*/+ animals from homozygous BALB *Soc* congenic lines than from *Clock^Δ19*/+ littermates lacking the BALB allele (*Figure 1F*). The effect of *Soc* was significant on *Clock^Δ19*/+ mutant, but not on wildtype or homozygous *Clock^Δ19* backgrounds. The congenic analysis demonstrates that *Soc* is physically located within the 165.1–190.45-Mb portion of mouse chromosome 1 and thus defines a genomic interval containing a new clock-related gene.

### Interval specific SNP haplotype analysis of the *Soc* locus

Because classical inbred laboratory mouse strains are derived from a limited number of progenitor species and the genomes of inbred strains are an admixture of different domesticated stocks (*Wade et al., 2002*; *Frazer et al., 2007*; *Yang et al., 2011*), polymorphisms at the *Soc* locus may be ancestral. If so, other inbred strains that share identity by descent with the *Soc* locus should also suppress *Clock*. To test this hypothesis, we characterized 14 additional laboratory inbred strains by crossing to B6 *Clock^Δ19* mice to create F1 hybrids. Further, because strain background can affect wild-type circadian free-running period, a two-way ANOVA is required to distinguish the effects of the strain background (evident in wild-type) from the effects of the strain background on the expression of the *Clock* mutation (evident as a strain-by-genotype interaction effect). If we detected a significant strain-by-genotype interaction, we accepted this as evidence of suppression of the *Clock* mutation. Of the 14 inbred strains tested, we identified seven additional suppressor and seven non-suppressor inbred strains (*Figure 2A*; *Table 1*). These results suggest that the *Soc* phenotype occurs as a consequence of shared ancestral alleles.

We performed pairwise SNP allele comparisons between B6 and the other 15 inbred strains in the 30-Mb interval from 160 to 190 Mb of mouse chromosome 1 using a total of 2714 SNPs (*Figure 2B*). Alternating regions of low and high SNP diversity are apparent in which low variation intervals represent strains sharing identity by descent, compared to high variation intervals that represent divergent ancestry. Within the 30-Mb interval, there is a region of high sequence variation between B6 and the suppressor strains, but identical by descent between B6 and the non-suppressor strains (*Figure 2B*). This region (green shading) spans 900 kb and contains 22 transcription units (*Figure 2C*; *Figure 2—source data 1*). Using the Mouse Phylogeny Viewer (*Yang et al., 2011*), the imputed subspecific origin of this 900 kb region reveals two sets of haplotypes that arise from *Mus domesticus*

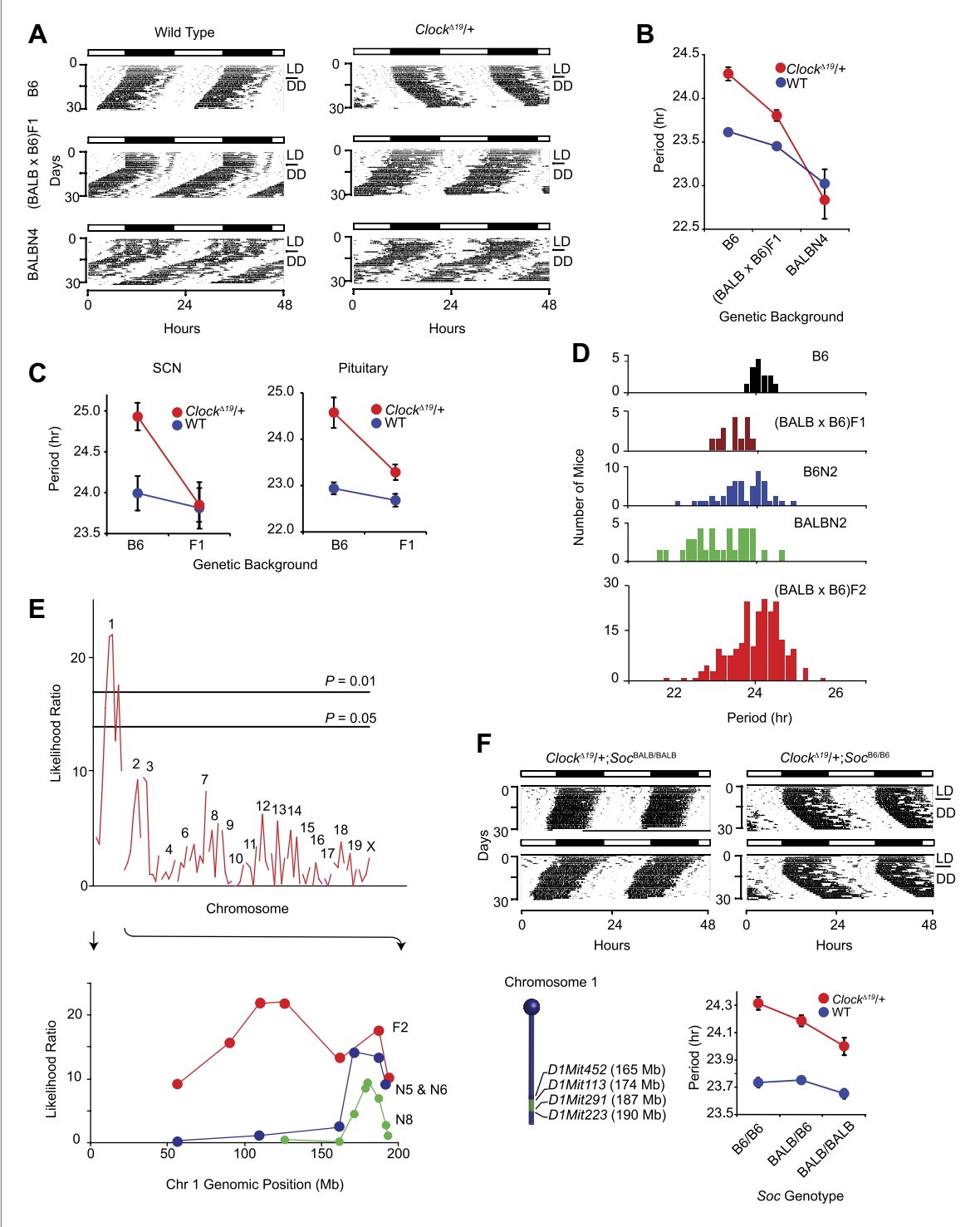

**Figure 1**. Suppression of *Clock$^{\Delta 19}$/+* circadian phenotype on the BALB genetic background. (**A**) Representative locomotor activity records of wild-type (left) and *Clock$^{\Delta 19}$/+* (right) animals. The top two records are from B6 coisogenic mice, the middle two are from (BALB x B6)F1 hybrids, and the bottom two are from ([BALB x B6]F1 x BALB)N4 mice. (**B**) Effect of genetic background on the free-running period of the locomotor activity rhythm. A two-way ANOVA was highly significant for *Clock$^{\Delta 19}$* genotype (p<10$^{-6}$), strain background (p<10$^{-8}$) and their interaction (p<10$^{-6}$). Each data point represents the mean ± SEM from 4–23 mice. (**C**) Effect of genetic background on the free-running period of PER2::LUC bioluminescence rhythms from *Clock$^{\Delta 19}$/+* SCN (left) and pituitary (right) explants. For SCN, a two-way ANOVA was significant for strain background (p<0.006) and interaction between *Clock$^{\Delta 19}$* genotype and strain background (p=0.02). The effect of *Clock$^{\Delta 19}$* genotype was not significant (p=0.11). For pituitary, a two-way ANOVA was significant for *Clock$^{\Delta 19}$* genotype (p<0.001), strain background (p<0.001), and their interaction (p=0.01). Each data point represents the mean ± SEM from 11–18 mice. *Figure 1. Continued on next page*

Figure 1. Continued

(**D**) Analysis of circadian period in hybrid populations between B6 and BALB animals. Numbers of animals used for period distribution analysis in each population include: B6 (*n* = 13), (BALB x B6)F1 (*n* = 12), ([BALB x B6]F1 x B6)N2 (*n* = 45), ([BALB x B6]F1 x BALB)N2 (*n* = 31), and (BALB x B6)F2 (*n* = 222). (**E**) Mapping of the *Soc* locus. A significant association was detected with markers at the distal end of chromosome 1 in a (BALB x B6)F2 cross (*n* = 222). Horizontal lines indicate thresholds determined by 10,000 permutation tests. Higher resolution mapping of the *Soc* locus was accomplished by selecting nearly isogenic animals for phenotype and genotype (bottom). We performed association analysis with SSLP markers on chromosome 1 on the N5 and N6 combined (*n* = 41), and N8 (*n* = 193) generations. (**F**) Representative activity records of *Clock*$^{\Delta 19}$/+ animals from congenic mouse strains (top panel). The two left panels show activity records from B6 *Clock*$^{\Delta 19}$/+ mice carrying ~25 Mb of BALB chromosome 1. The two right panels are from littermate controls which do not carry the BALB *Soc* region of chromosome 1. A diagram of the congenic region (green) on chromosome 1 is show in the left bottom panel. A two-way ANOVA was highly significant for the *Clock*$^{\Delta 19}$ genotype (p<10$^{-8}$) and congenic genotype (p=4.2 × 10$^{-4}$). The interaction between the *Clock*$^{\Delta 19}$ and congenic genotypes was not significant (p=0.057). Each data point represents the mean ± SEM from 11–29 mice.

and *M. musculus*, respectively (*Table 2*). The suppressor strains all carry haplotypes originating from *M. domesticus* and the non-suppressor strains all carry haplotypes originating from *M. musculus*—thus confirming our hypothesis of an ancestral allele.

## Identification of *Soc*

It is well established that circadian clocks exist in the SCN and throughout the body (*Yoo et al., 2004*). Because the *Clock* suppressor phenotype occurs in both SCN and pituitary tissue explants (*Figure 1C*), the gene encoded by the *Soc* locus should also be expressed in both central and peripheral tissues. To search for co-expression, we profiled the expression of the 22 *Soc* candidates as well as 9 circadian clock genes in 10 different tissues. Via cluster analysis, we found that all nine clock genes expressed a similar pattern among the tissues examined (*Figure 3A*), but that only seven *Soc* candidates (*Vangl2*, *Usf1*, *Dcaf8*, *Copa*, *Pex19*, *Pea15,* and *Ncstn*) expressed a similar pattern to the clock genes. We thus prioritized these seven *Soc* candidate genes for further analysis.

The *Clock*$^{\Delta 19}$ mutation results from an A→T transversion in a splice donor site that causes exon skipping and disruption of the C-terminal transactivation domain of CLOCK (*King et al., 1997*). Consequently, the levels of *Per* and *Cry* mRNA are much lower in *Clock*$^{\Delta 19}$ mutants than in wild-type mice (*Lowrey and Takahashi, 2011*). Given the nature of the suppressor phenotype, we hypothesized that the product of the gene responsible for the *Soc* QTL should activate transcription from E-boxes in a manner similar to CLOCK:BMAL1. We tested E-boxes from *Per1* and *Per2* for activation by the seven *Soc* candidates in HEK293T cells. Of these, only *Usf1* significantly activated transcription from both *Per1* and *Per2* E-boxes (*Figure 3B*). We observed a clear dose response of *Per* promoter activation by USF1 (*Figure 3B*). Unlike CLOCK:BMAL1, activation of E-boxes by USF1 was not inhibited by the CRYs (*Figure 3B*). Although it has been reported previously that USF1 binds E-box sequences (*Ferré-D'Amaré et al., 1994*), our finding that, of the seven *Soc* candidates, only USF1 effects significant transactivation from circadian-relevant E-boxes, provided initial molecular evidence that *Usf1* is a candidate for the *Soc* locus.

We next tested the circadian expression of the seven candidate genes in liver tissue collected from *Clock*$^{\Delta 19}$/+ (BALB x B6)F1 and B6 animals every 4 hr on the third day of DD exposure. Among the genes tested, only the *Usf1* transcript exhibited significant up-regulation in F1 mice (*Figure 3C*). By western blot analysis, USF1 protein levels in the liver were ~40% higher in BALB animals (*Figure 3D*). To test whether the suppressor strain activates E-box-containing circadian clock genes in vivo, we examined the effect of strain background (B6 or [BALB x B6]F1) on expression levels of *Per1*, *Per2*, *Cry1* and *Cry2* in liver using quantitative PCR (qPCR). We detected significant up-regulation of all four circadian genes tested in the F1 background (*Figure 3E*), consistent with elevated expression of an E-box transcription factor such as USF1. Taken together these results strongly suggest that *Usf1* is a prime candidate gene for the *Soc* locus. Because there are no coding mutations in *Usf1* between the B6 and BALB mouse strains (*Figure 2—source data 1*), we hypothesized that the elevated expression of *Usf1* in the BALB suppressor strain is likely the cause of *Clock* suppression.

Given that *Soc* is a dominant suppressor and the candidate gene, *Usf1*, exhibits up-regulated expression in the BALB *Clock* suppressor strain, loss-of-function tests for *Usf1* are not appropriate. Instead, to test whether *Usf1* is the gene responsible for the *Soc* QTL, we created *Usf1* overexpression transgenic mice on an isogenic B6 genetic background. From previous work (*Hong et al., 2007*), we observed that a transgene fused to DNA fragments containing a CMV minimal promoter is induced

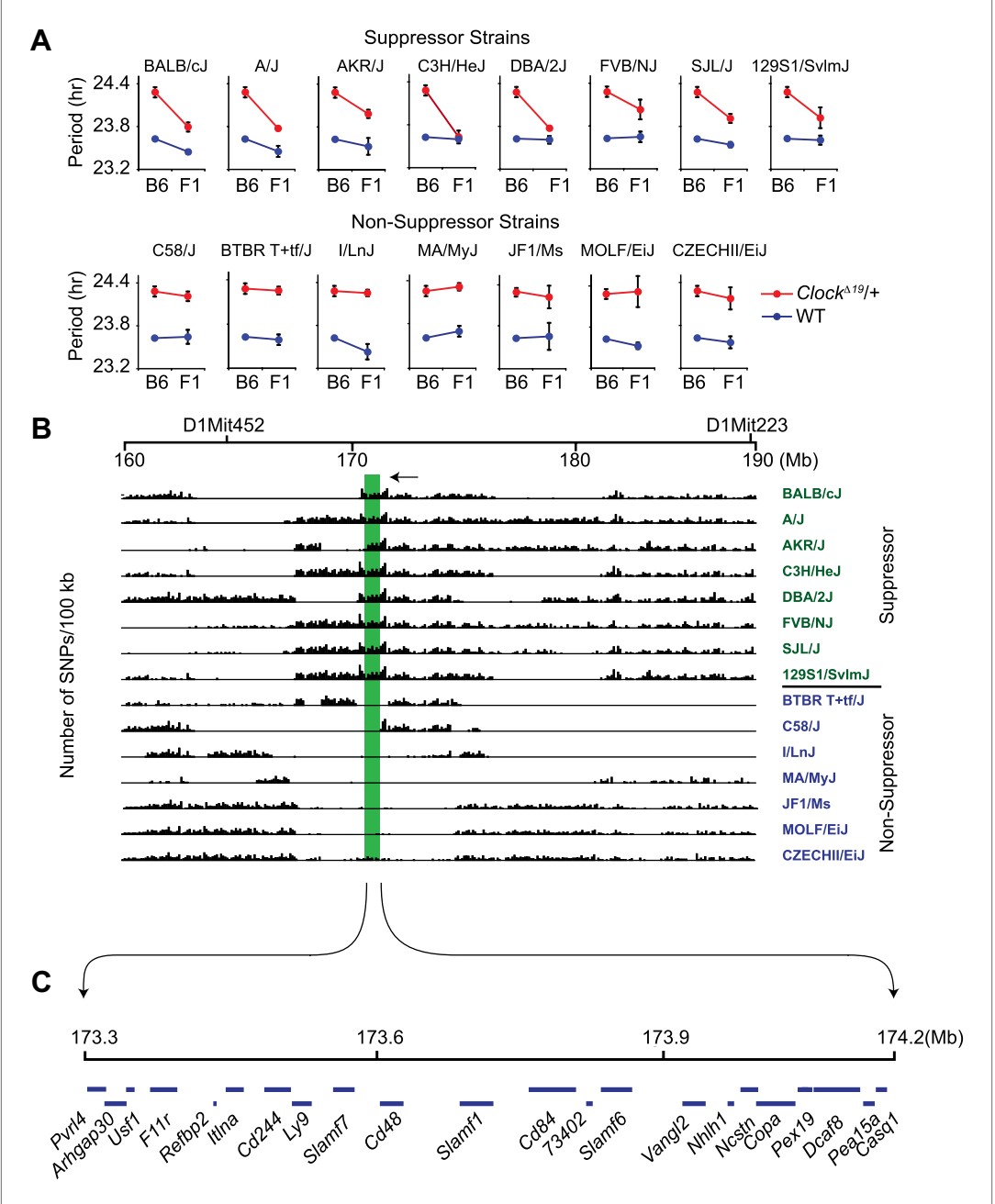

**Figure 2**. High-resolution mapping of the Soc locus using interval-specific SNP haplotype analysis. (**A**) Identification of $Clock^{\Delta19}$ suppressor and non-suppressor strains. B6 $Clock^{\Delta19}$/+ mice were crossed to 15 different inbred mouse strains. We used a two-way ANOVA to distinguish the effects of the strain background (evident in the wild-type mice) from the effects of the strain background on the expression of the $Clock^{\Delta19}$ mutation (evident as a strain-by-genotype interaction effect). Of the 15 inbred strains tested, we identified seven additional suppressor (A/J, AKR/J, C3H/HeJ, DBA/2J, FVB/NJ, SJL/J and 129S1/SvlmJ) and seven non-suppressor (C58/J, BTBR T+tf/J, I/LnJ, MA/MyJ, JF1/Ms, MOLF/EiJ, and CZECHII/EiJ) strains. A summary of the two-way ANOVA is provided (**Table 2**). In each panel the blue and red lines represent wild-type and $Clock^{\Delta19}$/+ animals, respectively; F1 indicates an F1 hybrid between B6 and the specific inbred strain indicated. Each data point represents the mean ± SEM from 4–22 mice. (**B**) Pairwise sequence comparison between B6 and 15 other mouse strains in the 100-kb BALB interval. We used 2714 SNPs within the 160–190 Mb interval of mouse chromosome 1. *Soc* should be located in regions of high sequence variation in suppressor strains (green) and regions of low sequence variation in non-suppressor strains (blue), relative to B6. The only region satisfying this criterion is indicated (green bar). (**C**) Physical map of the *Soc* interval. *Soc* maps to a 900-kb interval of chromosome 1. Blue bars in the top panel represent the 22 candidate genes identified within the interval.

The following source data are available for figure 2:

**Source data 1.** Single nucleotide polymorphisms (SNP) in the *Suppressor of Clock* (*Soc*) candidate region among 16 mouse strains.

**Table 1.** Two-way ANOVA for identification of Clock suppressor or non suppressor strains in *Figure 2A*

**Suppressor strain**

| Strain name | F-ratio (*Clock* × strain) | p value | df |
|---|---|---|---|
| BALB/cJ | 14.698 | <0.001 | df(1,61) |
| A/J | 13.13 | <0.001 | df(1,44) |
| AKR/J | 9.41 | 0.004 | df(1,44) |
| C3H/HeJ | 56.82 | <0.001 | df(1,52) |
| DBA/2J | 34.92 | <0.001 | df(1,46) |
| FVB/NJ | 7.24 | 0.010 | df(1,47) |
| SJL/J | 12.03 | 0.001 | df(1,55) |
| 129S1/SvImJ | 7.094 | 0.010 | df(1,67) |
| Non-suppressor strain | | | |
| C58/J | 0.65 | 0.423 | df(1,46) |
| BTBR_T+_tf/J | 0.01 | 0.911 | df(1,53) |
| I/LnJ | 2.72 | 0.107 | df(1,40) |
| MA/MyJ | 0.02 | 0.891 | df(1,52) |
| JF1/Ms | 0.49 | 0.490 | df(1,37) |
| MOLF/EiJ | 0.90 | 0.348 | df(1,37) |
| CZECHII/EiJ | 0.08 | 0.780 | df(1,46) |

Only the F-ratio for interaction is shown. Two-way ANOVA analyses for circadian period were performed with four groups; C57BL/6J WT, F1 WT, C57BL/6J Clock/+ and F1 Clock/+. Suppressor strains show a significant interaction between Clock genotype and strain background (Clock × strain).

at low levels, and thus created two independent *Usf1* transgenic lines, both of which exhibited a significant behavioral period-shortening effect in *Clock*$^{\Delta 19}$/+ animals, thereby minimizing the possibility of transgene position effects (*Figure 4A*). Quantitation of *Usf1* expression confirmed that the transgenic lines mimic the increase in *Usf1* levels observed in the BALB suppressor strain (*Figure 4B*). Based on these results, we conclude that *Usf1* is the gene encoded by the *Soc* locus. Consistent with the dominant action of *Soc*, *Usf1* knockout animals do not show changes in period length, but do exhibit a reduction in circadian amplitude and locomotor activity levels (*Figure 4C,D*).

For a gene to be a quantitative trait gene owing to an expression difference, the difference should be in a *cis*, rather than a *trans* regulatory element. We further examined the *Usf1* allele-specific expression difference between B6 and BALB mice in (B6 x BALB)F1 animals, which controls for trans-effects and environmental influences (*Cowles et al., 2002*). Using three independent methods (*Figure 5A,B,C*), we found that the *Usf1* expression difference between B6 and BALB is the result of polymorphisms in a *cis*-regulatory element causing allele-specific up-regulated expression of the BALB *Usf1* transcript.

To determine whether the allele-specific expression difference is the result of promoter polymorphisms between B6 and BALB, we tested basal promoter activity of ~2.3 kb upstream of the *Usf1* transcription start site using a luciferase reporter assay. *Usf1* promoter sequences from both B6 and BALB were cloned into the PGL4 luciferase reporter vector and transfected into HEK293T cells. Although both the B6 and BALB sequences activated the luciferase reporter, the BALB sequence resulted in a significantly higher signal than that from B6 (*Figure 5D*). By exchanging EcoRI-XhoI fragments between the B6 and BALB reporter clones, we narrowed the critical polymorphisms to the EcoRI-XhoI fragment (~1000 bp) and identified seven common polymorphisms (five SNPs and two deletions) that match the phenotypic distribution of 16 mouse strains (*Figure 5E*).

We next amplified ~100-bp fragments containing each of the seven polymorphisms from BALB and B6 genomic DNA and assayed their reporter gene activity to test whether the candidate SNPs affect promoter activity. Among the six fragments tested (SNPs 5 and 6 were contained within the same fragment), only two (SNPs 3 and 7) showed strong promoter activity and statistically higher activity from the BALB allele compared to the B6 allele (*Figure 5F*). SNPs 3 and 7 correspond to rs31538551 and rs31538547, respectively, in the NCBI SNP database. Next, we used site-directed mutagenesis on the B6 promoter (EcoRI-XhoI fragment) to introduce the corresponding BALB polymorphism at each of the two SNP loci and examined promoter activity. We observed a significant increase in luciferase signal by replacing the B6 allele of SNP7 (rs31538547) with the BALB allele (*Figure 5G*). This strongly suggests that SNP7 from the BALB *Usf1* promoter region is a quantitative trait nucleotide (QTN) responsible for the *Soc* phenotype.

## CLOCK:BMAL1 and USF binding to the E-box motif

Because an elevation of *Usf1* expression is most likely responsible for suppression of the *Clock*$^{\Delta 19}$/+ phenotype, we sought to determine the mechanism by which USF1 suppresses the *Clock* phenotype. Like CLOCK and BMAL1 (*Ripperger and Schibler, 2006*), USF1 binds to E-box motifs to regulate

**Table 2.** Subspecific origin of the Soc region of mouse chromosome 1

| Strain | Chromosome | Start position (bp) | End position (bp) | Subspecies | Sup or non-sup |
|---|---|---|---|---|---|
| BALB/cJ | 1 | 173300000 | 174200000 | Dom | S |
| A/J | 1 | 173300000 | 174200000 | Dom | S |
| AKR/J | 1 | 173300000 | 173302142 | Mus * | S |
| AKR/J | 1 | 173302461 | 174200000 | Dom | S |
| C3H/HeJ | 1 | 173300000 | 174200000 | Dom | S |
| DBA/2J | 1 | 173300000 | 174200000 | Dom | S |
| FVB/NJ | 1 | 173300000 | 174200000 | Dom | S |
| SJL/J | 1 | 173300000 | 174200000 | Dom | S |
| 129S1SvlmJ | 1 | 173300000 | 174200000 | Dom | S |
| C57BL/6J | 1 | 173300000 | 174200000 | Mus | NS |
| C58/J | 1 | 173300000 | 174028368 | Mus | NS |
| C58/J | 1 | 174187607 | 174200000 | Dom* | NS |
| BTBR T<+>tf/J | 1 | 173300000 | 174200000 | Mus | NS |
| I/LnJ | 1 | 173300000 | 174200000 | Mus | NS |
| JF1/Ms | 1 | 173300000 | 174200000 | Mus | NS |
| MOLF/EiJ | 1 | 173300000 | 174200000 | Mus | NS |
| CZECHII/EiJ | 1 | 173300000 | 174200000 | Mus | NS |

From the Mouse Phylogeny Viewer (**Yang et al., 2011**): http://msub.csbio.unc.edu. The AKR/J and C58/J strains contain the proximal and distal breakpoints, respectively, for the Soc locus, and thus there are two entries for these two strains. Dom = M. domesticus, Mus = M. musculus, S = suppressor, NS = non-suppressor.
*does not include Soc locus.

transcription (**Ferré-D'Amaré et al., 1994**). Using chromatin immunoprecipitation (ChIP) followed by qPCR analysis, we verified that CLOCK and USF1 can bind to common E-boxes in three circadian genes (*Dbp*, *Per2* and *Per1*) (**Figure 6A**). Although not as robust as CLOCK, we observed USF1 binding to four E-boxes (*Dbp* $E_P$, *Dbp* $E_{I2}$, *Per1* $E_{P1}$ and *Per1* $E_{P2}$). In *Clock$^{\Delta19}$* mutants, however, USF1 occupancy increases and CLOCK$^{\Delta19}$ occupancy decreases at these sites despite elevated CLOCK$^{\Delta19}$ mutant protein levels (**Figure 6B**; **Yoshitane et al., 2009**). Further, we observed a time-dependent DNA binding pattern of USF1 with a peak at night that is antiphase to that of CLOCK which peaks at ZT8 (**Figure 6C**).

To explore our observation that CLOCK$^{\Delta19}$:BMAL1 occupancy decreases in *Clock$^{\Delta19}$*/*Clock$^{\Delta19}$* mice we determined the relative affinity of CLOCK:BMAL1, CLOCK$^{\Delta19}$:BMAL1 and USF1 complexes to E-boxes using EMSA. Given our results demonstrating *Usf1* expression differences between the B6 (non-suppressor) and BALB (suppressor) mouse strains (**Figure 3**), we used isogenic B6 animals in these experiments to insure that USF1 levels were constant as we examined E-box occupancy of CLOCK, CLOCK$^{\Delta19}$, and USF1. We examined the same E-box sites in the *Dbp*, *Per1*, and *Per2* genes used for ChIP analysis (**Figure 6A**), and confirmed the identity of gel shift complexes by supershift using specific antibodies to CLOCK, BMAL1 and USF1 (**Figure 6D**). At all E-boxes, we detected three types of DNA:protein complexes: CLOCK:BMAL1 tandem heterodimers (CB2), CLOCK:BMAL1 heterodimers (CB1), and USF (consisting of USF1:USF1 and possibly USF1:USF2 or USF2:USF2 dimers). As shown previously, these E-box sites consist of tandem E-boxes spaced 6–10 nucleotides apart, and binding to these tandem sites appears to be cooperative (**Figure 6E**; **Rey et al., 2011**). Interestingly, wild-type CLOCK:BMAL1 complexes bind primarily as a tandem complex (CB2) at all four E-box sites with the *Dbp* $E_P$ and *Per2* E' E-boxes showing the most bias towards tandem binding (**Figure 6D**). By contrast, the mutant CLOCK$^{\Delta19}$:BMAL1 heterodimer binds primarily as a single complex (CB1) at all four sites. Antibody supershift experiments affected both the CB1 and CB2 complexes, consistent with the hypothesis that these complexes, whether containing mutant or wild-type CLOCK, represent single or tandem CLOCK:BMAL1 binding complexes, respectively (**Figure 6D**).

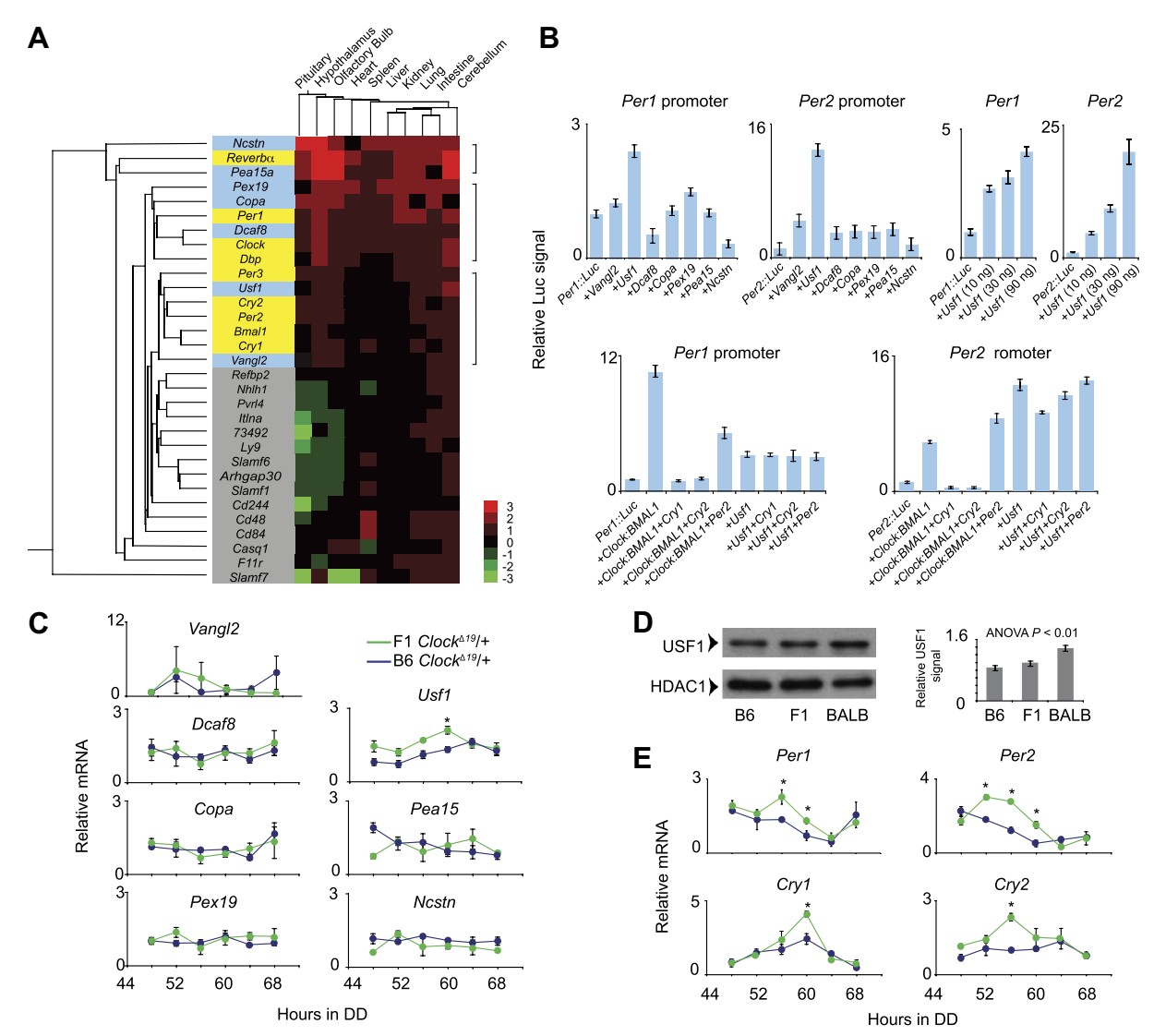

**Figure 3**. *Usf1* is a candidate for the *Soc* locus. (**A**) Gene expression profiling of 22 *Soc* candidates (gray and light blue) and nine clock genes (yellow) in 10 different tissues from *Soc* congenic *Clock*$^{\Delta 19}$/+ mice. All tissues were collected at ZT6. The RNA copy number was calculated by a method described previously (***Uno and Ueda, 2007***). Copy number within all tissues for all analyzed genes was normalized (Z = mean/SD). We coded Z scores in seven different colors. Cluster analysis was performed using Systat Software. Only seven of the 22 *Soc* candidates (*Vangl2, Usf1, Dcaf8, Copa, Pex19, Pea15* and *Ncstn*) show a similar expression profile with that of the clock genes within the 10 tissues examined. (**B**) USF1 activates *Per1* and *Per2* promoters. The seven *Soc* candidates were tested for *Per1* and *Per2* promoter activation (top left panels). Only USF1 activates both promoters. Each data point represents the mean ± SEM from three replicates. Activation of both *Per1* and *Per2* promoters by USF1 occurs in a dose-dependent manner (top right panels). USF1-mediated transcription is not inhibited by the CRY proteins. While there is strong negative inhibition by CRY of CLOCK:BMAL1-mediated transcription from the *Per* promoter, *Per* promoter activation by USF1 is not inhibited by either CRY1 or CRY2 (bottom panels). (**C**) Upregulation of *Usf1* mRNA in liver tissue of *Clock*$^{\Delta 19}$ suppressor background animals. Liver tissue was collected every 4 hr following 2 days of DD exposure. Among the seven *Soc* candidates tested, only *Usf1* was increased in liver from F1 *Clock*$^{\Delta 19}$/+ animals as determined by a two-way ANOVA (p=0.006). We did not detect an effect of time on *Usf1* expression. In each panel, lines represent B6 *Clock*$^{\Delta 19}$/+ (blue) and (BALB x B6)F1 *Clock*$^{\Delta 19}$/+ (green) animals. Each data point represents the mean ± SEM from 2–4 mice. Asterisks indicate significant differences between *Clock*$^{\Delta 19}$ suppressor and non-suppressor strain values at each time point (Tukey's post hoc, p<0.05). (**D**) Upregulation of USF1 protein in the liver nuclear extract from BALB/cJ genetic background. The left panel shows western blots for USF1 from B6, (BALB x B6)F1 and BALB male mouse liver samples. HDAC1 was used as a loading control. Mouse nuclear extract was prepared at ZT8. The right panel shows the normalized value for USF1 signal against HDAC1. Each bar represents the mean ± SEM from three replicates. A one-way ANOVA was significant for strain background (p=0.002). (**E**) Upregulation of E-box-containing circadian genes in liver from *Clock*$^{\Delta 19}$ suppressor background animals. We quantified *Per1, Per2, Cry1* and *Cry2* mRNA levels in the same liver samples used in (**C**). By a two-way ANOVA, we detected significant upregulation in *Per1* (p=1.5 × 10$^{-5}$), *Per2* (p<10$^{-8}$), *Cry1* (p=1.0 × 10$^{-8}$) and *Cry2* (p=6.6 × 10$^{-3}$) transcripts. Asterisks indicate significant differences between *Clock*$^{\Delta 19}$ suppressor and non-suppressor strain values at each time point (Tukey's post hoc, p<0.05).

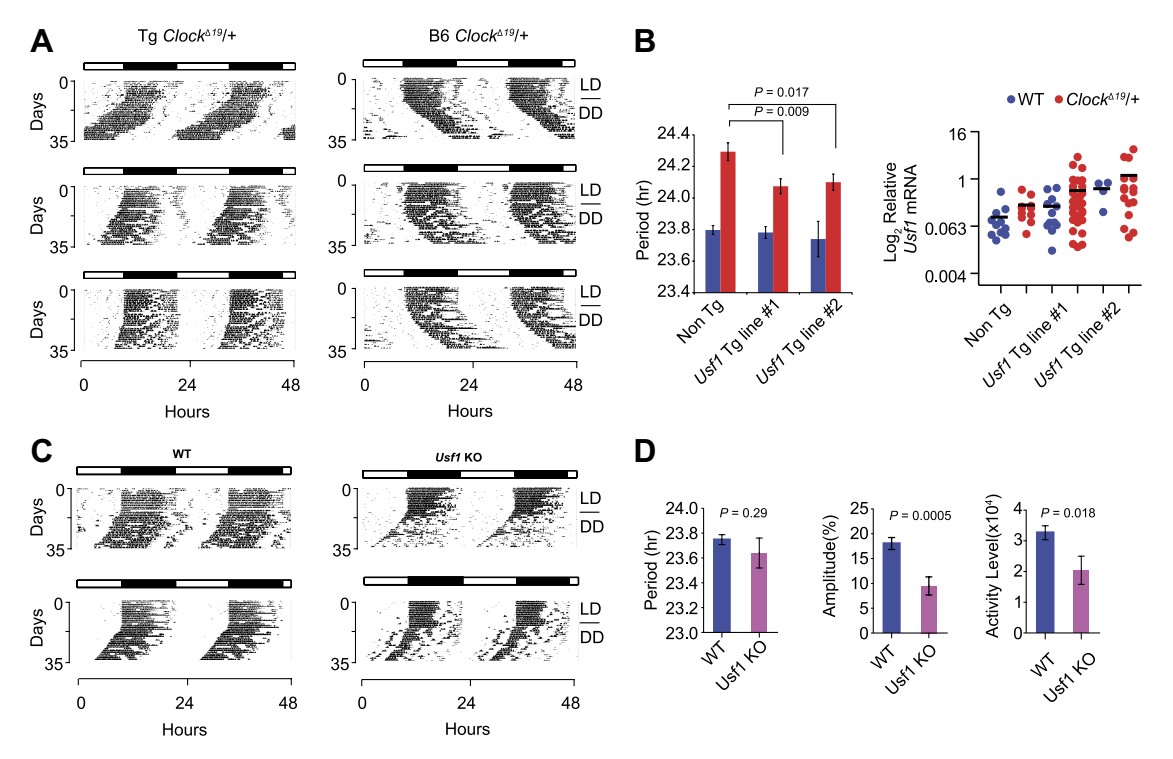

**Figure 4**. Transgenic expression of *Usf1* suppresses *Clock*$^{\Delta 19}$/+ mice and mimics *Soc*. (**A**) Representative locomotor activity records of *Usf1* transgenic *Clock*$^{\Delta 19}$/+ mice. Records on the left panel are from *Clock*$^{\Delta 19}$/+ mice carrying a *Usf1* transgene while those on the right are from *Clock*$^{\Delta 19}$/+ controls. All mice are C57BL/6J isogenic. (**B**) Analysis of circadian behavior in *Usf1* transgenic *Clock*$^{\Delta 19}$/+ mice. Significant period shortening was detected in two *Usf1* transgenic lines in *Clock*$^{\Delta 19}$/+ (red), but not wild-type (blue) mice. Each bar represents 5–35 mice from the C57BL/6J isogenic background (left). The right panel shows level of *Usf1* mRNA in hypothalamus at ZT6. Blue dots represent WT and red dots represent *Clock*$^{\Delta 19}$/+. A two-way ANOVA was significant for transgene (Tg) genotype (p=0.006). The effect of *Clock* genotype was not significant (p=0.18). There was no significant interaction between Tg and *Clock* genotype (p=0.616). (**C**) Circadian activity rhythms in *Usf1* knockout mice. Representative activity records of two wild type (left) and two *Usf1* knockout (right) mice. (**D**) Circadian period was not different between WT and *Usf1* KO mice. However, circadian amplitude and daily activity in constant darkness were significantly lower in *Usf1* KO mice. Data represent the mean ± SEM from 26 WT and 11 *Usf1* KO mice.

To measure the affinity of native CLOCK:BMAL1 protein-DNA interactions, we performed satura-tion binding experiments using EMSA to determine the apparent $k_d$ and $B_{max}$ values for the different complexes and E-box sites. To estimate the binding affinity of CLOCK:BMAL1, CLOCK$^{\Delta 19}$:BMAL1, and USF, liver nuclear extracts from ZT8 were incubated with increasing amounts of $^{32}$P labeled *Dbp* $E_P$, *Dbp* $E_{I2}$, *Per2* E' or *Per1* $E_{P1}$ E-box double-stranded oligonucleotide probes (**Figure 7A**). The binding of each of the complexes at the four E-boxes are presented as both absolute and relative binding plots to show both the relative amount of binding as well as the relative affinity (**Figure 7B**). At low probe concentrations, the wild-type CLOCK:BMAL1 complex (CB2) binds more strongly than CLOCK$^{\Delta 19}$:BMAL1 (CB1)—demonstrating the higher affinity of the wild-type complex for DNA. Although the affinity varies depending on the specific site, $K_d$ values of wild-type CLOCK:BMAL1 range from 0.86 nM for *Dbp* $E_{I2}$ to 19 nM for *Per1* $E_{P1}$, while the $K_d$ of CLOCK$^{\Delta 19}$:BMAL1 ranges from 9 to 48 nM for *Dbp* $E_{I2}$ and *Per1* $E_{P1}$, respectively (**Table 2**). $K_d$ estimates indicate a binding affinity change from 2.5- to 11.8-fold between wild-type and mutant CLOCK:BMAL1 complexes depending on the E-box site (**Figure 7B**; **Table 3**). Parenthetically, these data also show that the affinity and maximal binding of CLOCK:BMAL1 vary widely among native E-box sites, thus it is important to base interpretations on more than one E-box site. There is no significant difference in USF1 binding affinity between the two genotypes and, in fact, the $K_d$ of USF1 is similar to the mutant CLOCK$^{\Delta 19}$:BMAL1 complex (**Figure 7B**; **Table 3**).

In addition to these overall trends, specific E-box sites also provide further insight into the affinity of CLOCK:BMAL1 binding. In the case of the *Dbp* $E_{I2}$ site (**Figure 7A**), where wild-type CLOCK:BMAL1

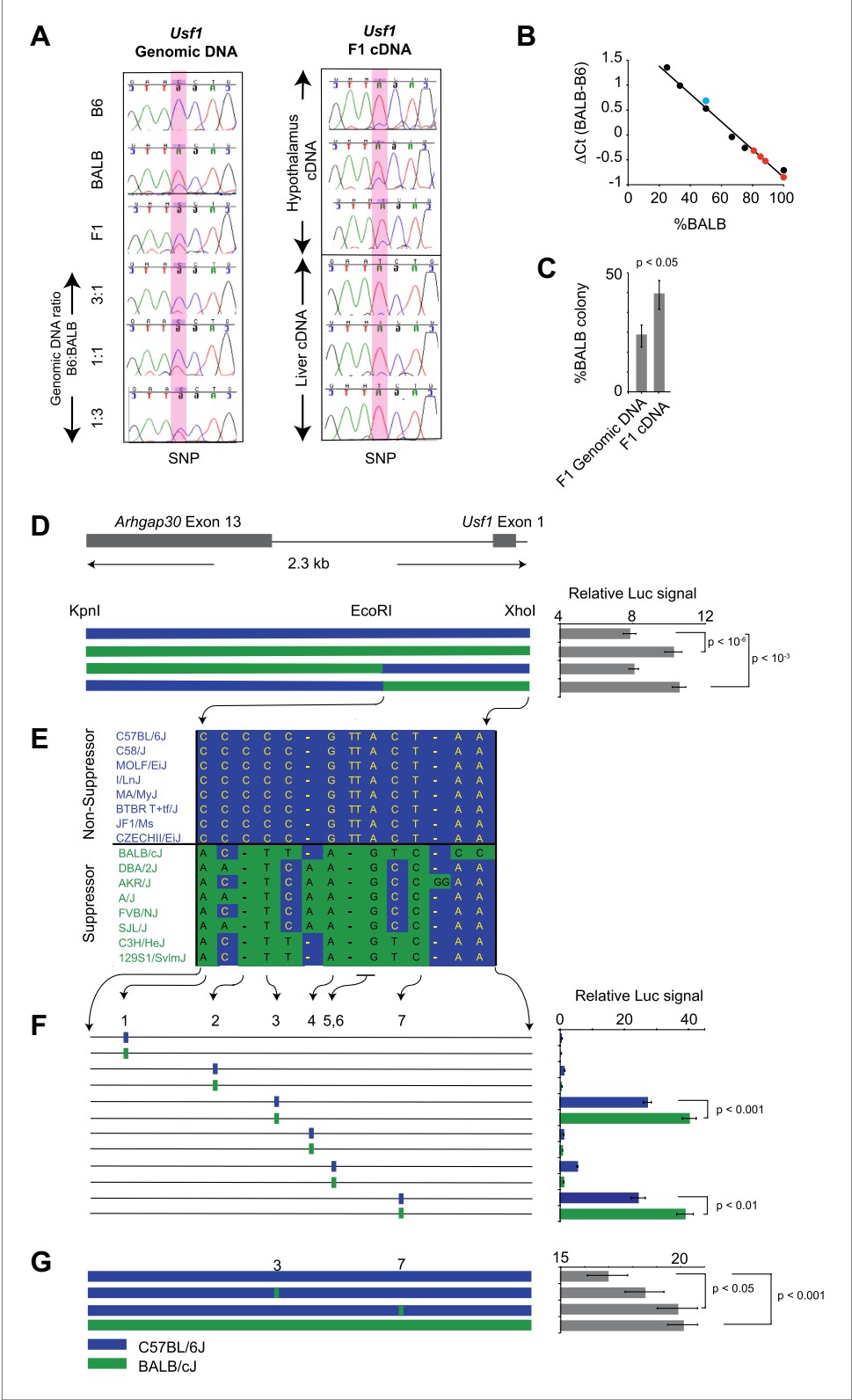

**Figure 5**. *Usf1* promoter analysis. (**A**) Allele-specific expression difference between B6 and BALB *Usf1* mRNA revealed by DNA sequencing. We chose an SNP at the 3'-UTR of *Usf1* as a marker to distinguish between the B6 and BALB alleles. The amplicon size from genomic DNA and cDNA are identical. We PCR-amplified the SNP-containing region in genomic DNA samples from B6, BALB, (B6 x BALB)F1 and three samples containing

*Figure 5. Continued on next page*

*Figure 5. Continued*

different mix ratios between B6 and BALB DNA (3:1, 1:1 and 1:3, respectively) as shown in the left panel. The panel on the right shows (B6 x BALB)F1 *Usf1* cDNA sequence from hypothalamus (top three) and liver (lower three). The PCR products were sequenced by an ABI3700 machine to compare the fluorescent signal peak ratio at the SNP locus as an indicator of the copy number ratio of the two alleles. This clearly demonstrates that the expression difference of *Usf1* between B6 and BALB is the result of polymorphisms in *cis* regulatory elements. (**B**) Analysis of allele-specific expression differences between B6 and BALB *Usf1* mRNA by quantitative PCR. We amplified genomic DNA mixtures described in (**A**) with allele-specific PCR primers and created a standard curve for %BALB genomic DNA vs $\Delta C_t$ (black data points). The blue data point is from F1 genomic DNA, which is naturally a 1:1 mixture of B6 and BALB. Red data points represent F1 cDNA from hypothalamus. Based on the standard curve generated from genomic DNA, 80% of the F1 cDNA contains the BALB *Usf1* allele. (**C**) Analysis of allele-specific expression differences between B6 and BALB *Usf1* mRNA by cloning PCR products. We amplified both F1 genomic DNA and cDNA using primers flanking an SNP in exon 10 (rs31093636) of *Usf1*. We generated four independent PCR products from a single F1 genomic DNA sample and four PCR products from four different F1 cDNA samples. The exon 10 SNP creates a restriction fragment length polymorphism such that only the BALB allele is cleaved by restriction enzyme TfiI. Each PCR product was cloned into a TA plasmid vector. From each transformation, we picked 24–48 colonies. We isolated 143 colonies containing F1 genomic DNA and 172 colonies containing F1 cDNA. This analysis, like that in (**B**), demonstrates that the F1 cDNA contains a higher percentage of the BALB *Usf1* allele than expected. (**D**) *Usf1* promoter analysis between B6 and BALB mouse strains. The putative *Usf1* promoter sequence (~2.3 kb upstream of exon 1) was cloned into the pGL4 luciferase reporter vector and 90 ng of either B6 or BALB construct was transfected into HEK293T cells. Promoter activity from the BALB clone is significantly higher than that from the B6 clone ($p<10^{-6}$). Next, we swapped EcoRI-XhoI fragments between the B6 and BALB constructs. The B6 clone containing the BALB EcoRI-XhoI fragment has significantly higher activity than the original B6 clone ($p<10^{-3}$). The activity is equivalent to the original BALB clone. On the other hand, the BALB clone containing the B6 EcoRI-XhoI fragment exhibits essentially similar activity as the B6 intact clone. The blue and green bars indicate B6 and BALB DNA fragments, respectively. Each data point represents mean ± SEM of 18–36 samples. (**E**) SNP distribution pattern among 16 mouse strains in the *Usf1* promoter candidate region. We detected 14 polymorphisms among 16 mouse strains within the ~1000 bp candidate region. The top eight strains are *Clock*$^{\Delta 19}$ non-suppressor strains and the bottom eight are *Clock*$^{\Delta 19}$ suppressor strains. There are only seven polymorphisms that perfectly match the phenotype distribution pattern. Blue indicates the B6 allele and green indicates the BALB allele. (**F**) Putative *Usf1* promoter SNPs (containing ~100 bp of flanking sequence of SNP1, SNP2, SNP3, SNP4, SNP5&6 and SNP7) from B6 and BALB were cloned into the pGL4 luciferase reporter vector. Only SNP3 and SNP7 show elevated promoter activity. In both cases, the BALB allele has significantly higher activity ($p<0.001$) than the B6 allele ($p<0.01$). In the NCBI RefSNP database, SNP3 corresponds to rs31538551 and SNP7 to rs31538547. Each data point represents the mean ± SEM of six samples. (**G**) We mutagenized the B6 promoter EcoRI-XhoI fragment to introduce the BALB allele at either SNP3 or SNP7. We observed a significant increase in the luciferase signal by SNP7 ($p<0.05$). Although we did not detect significant upregulation by the BALB allele at SNP3 ($p=0.18$), the level of luciferase was elevated with the BALB allele compared to the B6 allele. Each data point represents the mean ± SEM of 12 samples.

binds both as single and tandem complexes, the tandem CB2 site has much higher affinity than CB1, consistent with the prediction of previous work (*Rey et al., 2011*). In contrast, at the *Dbp* $E_P$ site (*Figure 7A*), where mutant CLOCK$^{\Delta 19}$:BMAL1 complex can be seen binding as both single CB1 and tandem CB2 complexes, the affinities of the two mutant complexes are similar and lower than wild-type CLOCK:BMAL1 complex. This suggests that the affinity of the mutant CLOCK$^{\Delta 19}$:BMAL1 complex is lower for two reasons: first, it does not preferentially bind as a tandem complex, thus binding may not be cooperative; and second, its affinity, even as a tandem complex, remains lower suggesting that the *Clock*$^{\Delta 19}$ mutation interferes both with cooperativity and with affinity. Thus, our saturation binding experiments clearly demonstrate that the affinity of CLOCK$^{\Delta 19}$:BMAL1 is significantly lower relative to the wild-type complex, and that this lower affinity is similar to the affinity of USF1 at the same sites. This suggests that under normal conditions, the wild-type CLOCK:BMAL1 complex binds with much higher affinity than the USF1 complex, but when mutant, the affinities of the CLOCK$^{\Delta 19}$:BMAL1 and USF1 complexes are comparable which allows USF1 to bind more effectively to E-box sites.

## Genome-wide location analysis of CLOCK:BMAL1 and USF1 binding sites

Because USF1 and CLOCK:BMAL1 compete for binding at the same E-boxes, and because CLOCK$^{\Delta 19}$:BMAL1 exhibits a lower affinity for these sites, we predicted that there should be a global

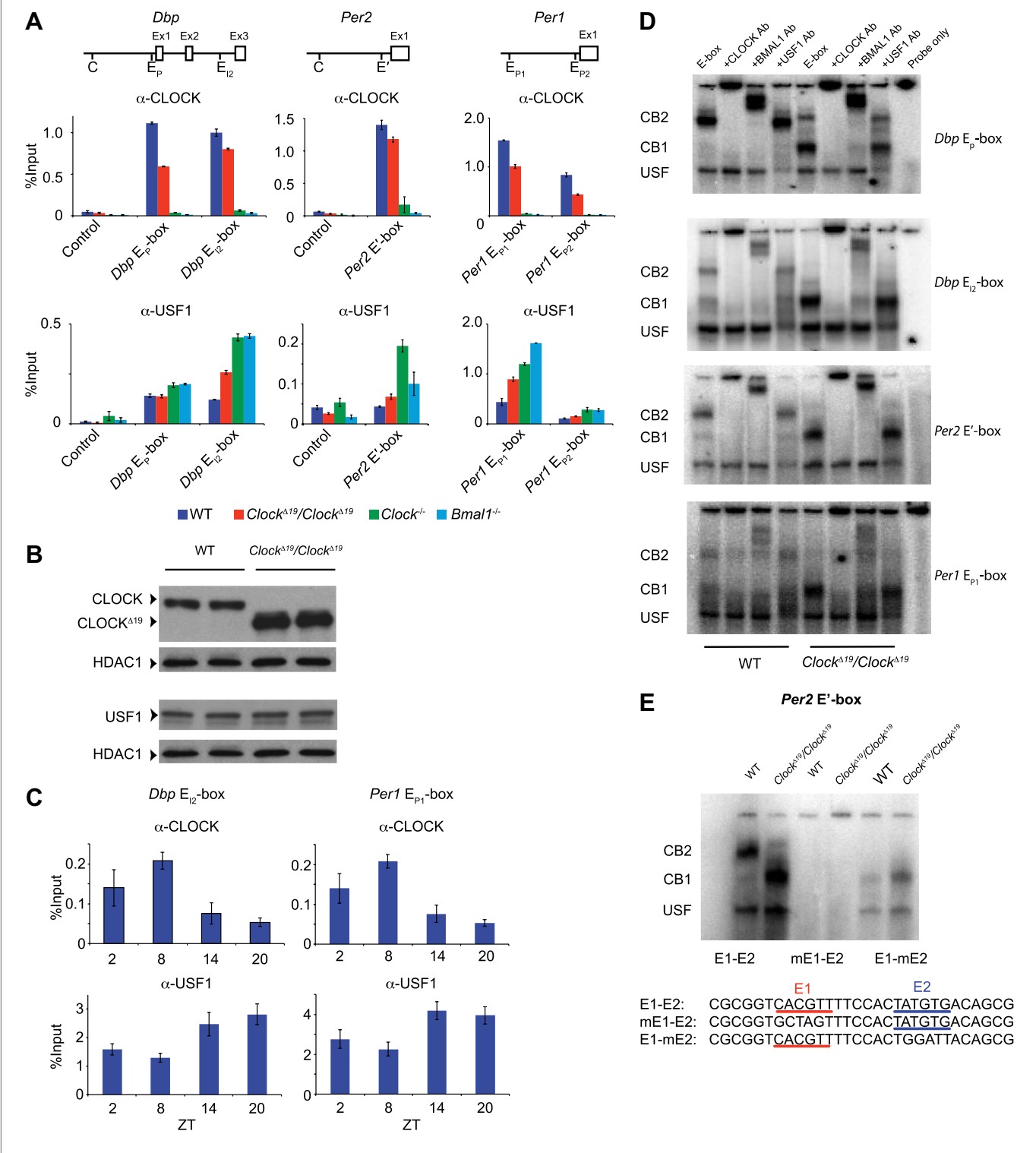

**Figure 6**. CLOCK:BMAL1 and USF1 binding to the E-box motif. (**A**) CLOCK and USF1 binding at *Dbp*, *Per2* and *Per1* E-boxes revealed by ChIP-qPCR. E-box occupancy demonstrated by CLOCK (top panels) and USF1 (bottom panels) is shown. At all E-boxes examined, CLOCK binding is lower in *Clock*$^{\Delta 19}$/*Clock*$^{\Delta 19}$ animals compared to wild-type controls. In contrast, USF1 binding is elevated in *Clock*$^{\Delta 19}$/*Clock*$^{\Delta 19}$ mice compared to wild-type animals at the *Dbp* E$_{I2}$, *Per2* E' and *Per1* E$_{P1}$ and E$_{P2}$ E-boxes. USF1 binding is also elevated in *Clock* and *Bmal1* knockout mice. (**B**) Western blot for CLOCK and USF1 in liver nuclear extracts from wild-type and *Clock*$^{\Delta 19}$/*Clock*$^{\Delta 19}$ C57BL/6J isogenic animals. Tissues were harvested at ZT8. HDAC1 was used as a loading control. (**C**) Reciprocal pattern of CLOCK and USF1 binding at *Dbp* and *Per1* E-boxes with time of day using ChIP-qPCR. (**D**) Antibody validation

*Figure 6. Continued on next page*

*Figure 6. Continued*

of CLOCK:BMAL1 and USF1 complexes for EMSA experiments using tandem *Dbp* E$_P$-box , *Dbp* E$_{I2}$-box, *Per2* E'-box and *Per1* E$_{P1}$-box, top to bottom, respectively, in liver nuclear extracts from wild-type and *Clock*$^{Δ19}$/*Clock*$^{Δ19}$ C57BL/6J isogenic animals at ZT8. CB2 (CLOCK:BMAL1 tandem heterodimeric complex); CB1 (CLOCK:BMAL1 heterodimer); USF (USF1 or USF2 homodimer or USF1:USF2 heterodimer). CB1 and CB2 complexes are completely abolished or shifted by CLOCK or BMAL1 antibodies, respectively. USF complexes are reduced significantly by USF1 antibody, with the remaining complex presumed to be USF2. (**E**) EMSA analysis of E1 and E2 mutants in the tandem *Per2* E'-box. Mutation of the E1 site in the *Per2* E'-box completely blocks CLOCK:BMAL1 and USF complex binding. Mutation of the E2 site blocks the formation of tandem heterodimeric CB2 complexes in WT liver nuclear extracts and only CB1 complexes are detected in both WT and mutant extracts. This confirms the critical role of the E1 site for binding as well as the requirement for the E2 site for tandem complex formation as reported (*Rey et al., 2011*).

increase in USF1 occupancy at CLOCK:BMAL1 binding sites in *Clock*$^{Δ19}$ mutant mice. To test this hypothesis we performed genome-wide location analysis of USF1, CLOCK and BMAL1 using ChIP-Seq on liver nuclei collected at ZT8 from wild-type and *Clock*$^{Δ19}$/*Clock*$^{Δ19}$ mice. Again, we used isogenic B6 mice in order to avoid the complication of different levels of USF1 in suppressor strains. Analysis of USF1 and CLOCK:BMAL1 binding sites reveals extensive overlap in binding in wild-type mice, with 497 of 1885 USF1 peaks binding either CLOCK or BMAL1 (*Figure 8A*). As predicted for *Clock*$^{Δ19}$/*Clock*$^{Δ19}$ animals, the overlap between USF1 and CLOCK:BMAL1 binding sites increases significantly to 1916 sites, an almost fourfold increase compared to wild-type controls (*Figure 8A*). In *Clock*$^{Δ19}$/*Clock*$^{Δ19}$ mutant animals, USF1 co-occupies 38% of CLOCK:BMAL1 sites (1916/5072)—a dramatic increase from 14% (497/3412) observed in wild-type animals. Surprisingly, USF1 binding also increases globally in homozygous *Clock*$^{Δ19}$ animals, even at sites that do not bind CLOCK or BMAL1, from 1885 peaks in wild-type animals to 6091 peaks in *Clock*$^{Δ19}$/*Clock*$^{Δ19}$ mutant animals.

Heat map analysis of the 1916 sites that are co-occupied by both USF1 and CLOCK:BMAL1 in *Clock*$^{Δ19}$/*Clock*$^{Δ19}$ mutant animals reveals a dramatic increase in binding intensity of USF1 in mutant mice (*Figure 8B,C*). Not only do the number of USF1 peaks increase, but also the binding intensity at existing sites increases in mutant animals. Concomitantly, in the mutant animals there is a decline in CLOCK binding intensity at these 1916 common binding sites (*Figure 8B,C*). We quantitated the binding intensity of the top 10% of the binding sites for all three ChIP-Seq experiments and performed comparisons between wild-type and *Clock*$^{Δ19}$/*Clock*$^{Δ19}$ animals (*Figure 8C*). For USF1 and CLOCK there was a significant increase and decrease, respectively, in binding between wild-type and mutant animals, but no change in BMAL1 binding (*Figure 8C*). Motif analysis of each group of binding sites identified the canonical E-box binding site, CACGTG, as the most significant motif, indicating both USF1 and CLOCK:BMAL1 bind to the same consensus sequence in both wild-type and *Clock*$^{Δ19}$/*Clock*$^{Δ19}$ mice (*Figure 8D*).

UCSC genome browser profiles are shown for examples of four genes (*Figure 8E*). *Per1* has at least five E-boxes, however ChIP-Seq analysis reveals two major peaks approximately 5 kb upstream of the transcription start site as well as two minor peaks further downstream (*Figure 8E*). The peak height of USF1 binding increases dramatically in *Clock*$^{Δ19}$/*Clock*$^{Δ19}$ animals, and there is a slight reduction in CLOCK binding in mutant animals. Within the *Dbp* gene there are also multiple E-boxes, and USF1 binding increases in *Clock*$^{Δ19}$/*Clock*$^{Δ19}$ animals on the promoter, and intron two E-boxes. On the *Dbp* promoter CLOCK and BMAL1 binding decreases in the *Clock*$^{Δ19}$/*Clock*$^{Δ19}$ animals. A similar pattern is observed for other clock genes such as *Rev-erbα* which contains multiple E-boxes (*Figure 8E*) and other target genes such as *Ilf3* (*Figure 8E*).

We extended this analysis of ChIP-Seq data to 41 E-boxes in 15 'reference clock genes' that are thought to be core regulators of the circadian clock (*Rey et al., 2011*). We specifically analyzed these E-box sites in the two genotypes for USF1 binding changes. Of these 41 sites, 23 bound USF1 with a significant increase in USF1 binding in the *Clock*$^{Δ19}$/*Clock*$^{Δ19}$ animals (*Figure 8F,G*). Thus, USF1 binds to a majority of reference clock genes and its occupancy on these sites increases in the *Clock*$^{Δ19}$/*Clock*$^{Δ19}$ mice.

These data suggest an intriguing mechanism of rescue of the *Clock*$^{Δ19}$ heterozygous phenotype by USF1. In this model, USF1 expression is higher in BALB than in B6 strain owing to a promoter polymorphism. Our data show that USF1 can competitively bind to the same E-boxes in vivo as CLOCK:BMAL1. Furthermore quantitative analysis of binding kinetics indicates that the CLOCK$^{Δ19}$:BMAL1 complex has a much lower binding affinity for E-boxes than its wild-type counterpart. This lowered affinity combined

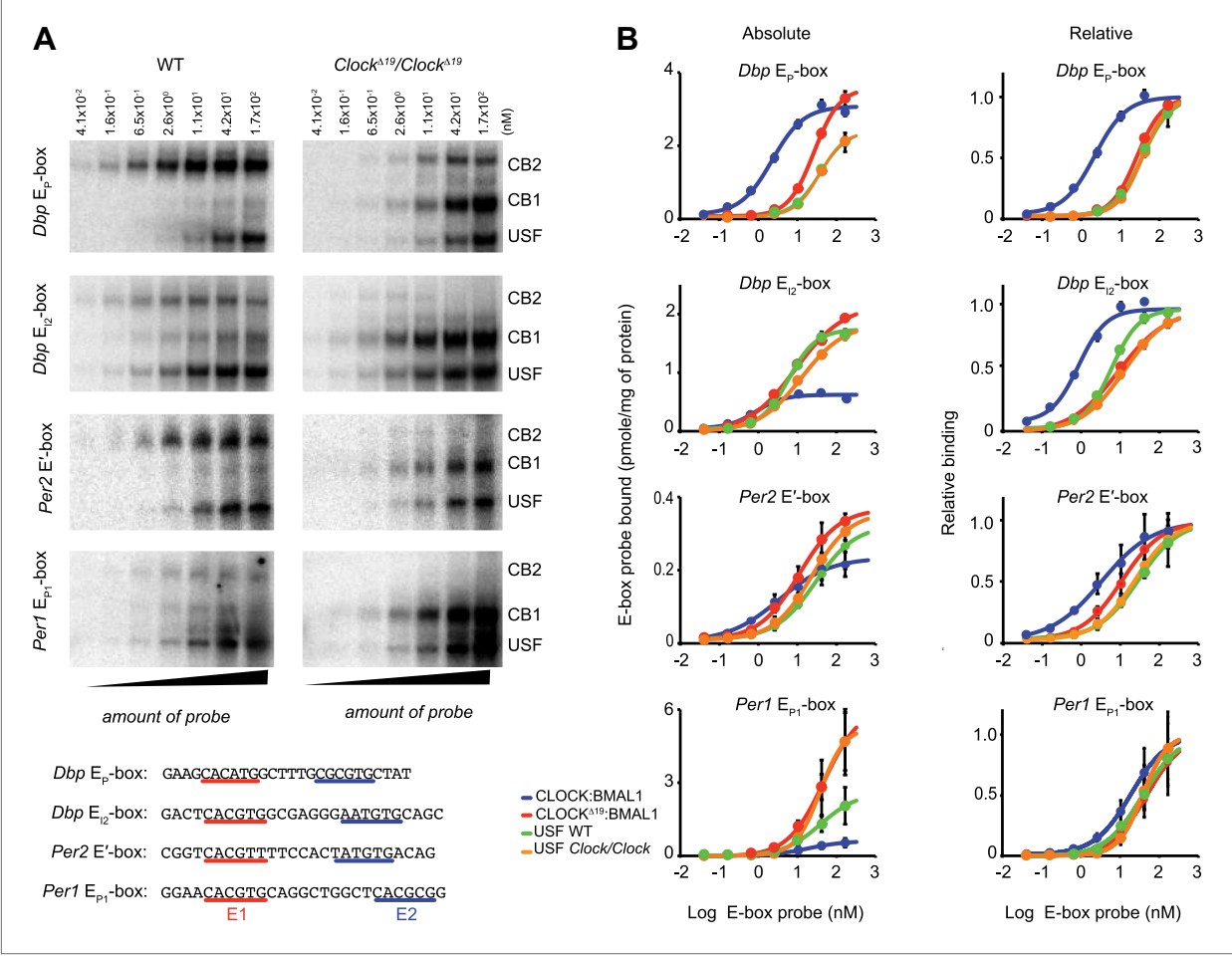

**Figure 7**. Native CLOCK:BMAL1 and USF1 binding affinity. (**A**) Saturation binding kinetics of CLOCK:BMAL1 and USF at E-boxes. EMSA analysis for the *Dbp* $E_P$, *Dbp* $E_{I2}$, *Per2* E', and *Per1* $E_{P1}$ E-boxes, top to bottom, respectively, with different probe concentrations. The left panels are from wild-type and the right panels are from *Clock*$^{Δ19}$/*Clock*$^{Δ19}$ animals. CB2 (CLOCK:BMAL1 tandem heterodimeric complex); CB1 (CLOCK:BMAL1 heterodimer); USF (USF1:USF1 or perhaps USF1:USF2, USF1:USF2). Sequences for the tandem E-boxes examined are given (bottom; E1 sequence, red; E2 sequence, blue). (**B**) Analysis of saturation binding kinetics as a function of probe concentration. Left panels represent absolute binding and right panels show binding relative to the maximum. At all sites, CLOCK$^{Δ19}$:BMAL1 binding affinity is lower than that for CLOCK:BMAL1 and similar to USF binding affinity. Each data point represents the mean ± SEM of three experiments.

with increased expression of USF1 in BALB mice accounts for the rescue of the *Clock*$^{Δ19}$/+ phenotype in this strain. We propose that USF1 acts as a partial agonist whose transcriptional regulation of circadian genes can compensate for the reduction in transcriptional activation by the CLOCK$^{Δ19}$:BMAL1 complex.

## Discussion

Taken together, we demonstrate that a dominant suppressor of the *Clock*$^{Δ19}$ mutation in the BALB genetic background is an ancestral allele of *Usf1* which carries a *cis*-regulatory SNP that enhances *Usf1* expression (*Figure 9A*). USF1 is ubiquitously expressed in both mouse and human and is a member of the evolutionarily conserved bHLH-Zip transcription factor family. USF1 participates in the regulation of several processes including lipid and carbohydrate metabolism, immune responses, and the cell cycle (*Corre and Galibert, 2005*). USF1 interacts with the circadian clock gene pathway by binding to E-box regulatory sites in common with those bound by CLOCK:BMAL1 to regulate circadian gene expression (*Figure 9B*). We find that the mutant CLOCK$^{Δ19}$:BMAL1 complex binds with much lower affinity than wild-type CLOCK:BMAL1 complexes, arising in part from the absence of cooperativity of binding to tandem E-box sites seen with the wild-type CLOCK:BMAL1 complex (*Figure 9C*). Because

**Table 3.** Apparent binding affinity (Kd) and maximal binding (Bmax) for CLOCK:BMAL1 and USF at four different E-boxes in *Figure 7*

|  | CLOCK:BMAL1 | CLOCK$^{\Delta19}$:BMAL1 | USF1 WT | USF1 *Clock$^{\Delta19}$/Clock$^{\Delta19}$* |
|---|---|---|---|---|
| *Dbp* E$_P$-box |  |  |  |  |
| Bmax | 0.97 ± 0.02 | 1.25 ± 0.05 * | 0.83 ± 0.06 | 0.87 ± 0.04 |
| Kd | 2.01 ± 0.22* | 32.94 ± 3.92 | 42.8 ± 9.30 | 49.56 ± 6.20 |
| *Dbp* E$_{I2}$-box |  |  |  |  |
| Bmax | 0.79 ± 0.02* | 2.38 ± 0.05 | 2.25 ± 0.05 | 2.10 ± 0.05 |
| Kd | 0.71 ± 0.10* | 6.18 ± 0.54 | 6.10 ± 0.52 | 9.64 ± 0.96 |
| *Per2* E'-box |  |  |  |  |
| Bmax | 0.26 ± 0.02* | 0.43 ± 0.02 | 0.36 ± 0.03 | 0.41 ± 0.02 |
| Kd | 2.00 ± 0.64* | 8.05 ± 1.57 | 17.81 ± 5.08 | 16.71 ± 2.46 |
| *Per1* E$_{P1}$-box |  |  |  |  |
| Bmax | 0.74 ± 0.12* | 7.38 ± 1.18 | 3.06 ± 0.64 | 7.96 ± 2.01 |
| Kd | 16.55 ± 9.49 | 43.69 ± 19.40 | 33.36 ± 20.99 | 58.62 ± 37.82 |

Values are the mean + SEM derived from four-parameter nonlinear curve-fitting.
*indicates that the Bmax or Kd is significantly different from other groups, p<0.0001.

USF1 binds as a dimer, the 40% increase in USF1 levels observed in the *Clock* suppressor strain could lead to a substantial increase in DNA binding if those levels are below the K$_d$ for USF1 binding. In addition, because the PER:CRY negative feedback complex cannot repress USF1-mediated trans-activation (*Figure 3B*), a modest increase in USF1 could activate target genes more effectively than CLOCK$^{\Delta19}$:BMAL1. Thus, USF1 can suppress the *Clock$^{\Delta19}$* mutant by increased occupancy of CLOCK:BMAL1 E-box sites, acting as a partial agonist for CLOCK:BMAL1-mediated transcription. What might the role of USF1 be in a *Clock* wild-type background? We have found that as CLOCK:BMAL1 DNA binding decreases at night (*Rey et al., 2011*; *Koike et al., 2012*), USF1 occupancy increases at these sites, exhibiting an antiphase temporal DNA binding pattern (*Figure 6C*). We speculate that the role of USF1 at CLOCK:BMAL1 sites could be to maintain an open chromatin state to facilitate CLOCK:BMAL1 binding on the following circadian cycle although additional work would be required to test this hypothesis. In addition, in *Usf1* knockout mice, we observe a reduction in circadian amplitude and activity levels showing that USF1 plays a role under normal conditions to enhance circadian rhythmicity and robustness. While it is tempting to speculate that these effects of *Usf1* might work at the level of the cell autonomous circadian oscillator, additional experiments would be required to determine at what level of organization (cell, circuit or higher) these changes in amplitude at the behavioral level originate.

In summary, the global interactions between CLOCK:BMAL1 and USF1 reveal an extensive and previously unknown interface linking these two transcriptional networks. It will be interesting in future work to determine whether the shared targets of these two pathways affect phenotypes beyond the circadian system. Taken together, these results show that USF1 is a significant modulator of molecular and behavioral circadian rhythms in mammals.

## Materials and methods

### Animals

All animals used in this study were raised at Northwestern University or the University of Texas Southwestern Medical Center in a 12 hr light/12 hr dark cycle (LD12:12) from birth. All animal care and use procedures were in accordance with guidelines of the Northwestern University and UT Southwestern Institutional Animal Care and Use Committees. At 8–12 weeks of age, mice were transferred to individual cages equipped with running wheels and housed in LD12:12 conditions. After a minimum of 7 days, animals were transferred to DD conditions for 3 weeks. The free running period was calculated using a $\chi^2$ periodogram as described previously (*Shimomura et al., 2001*).

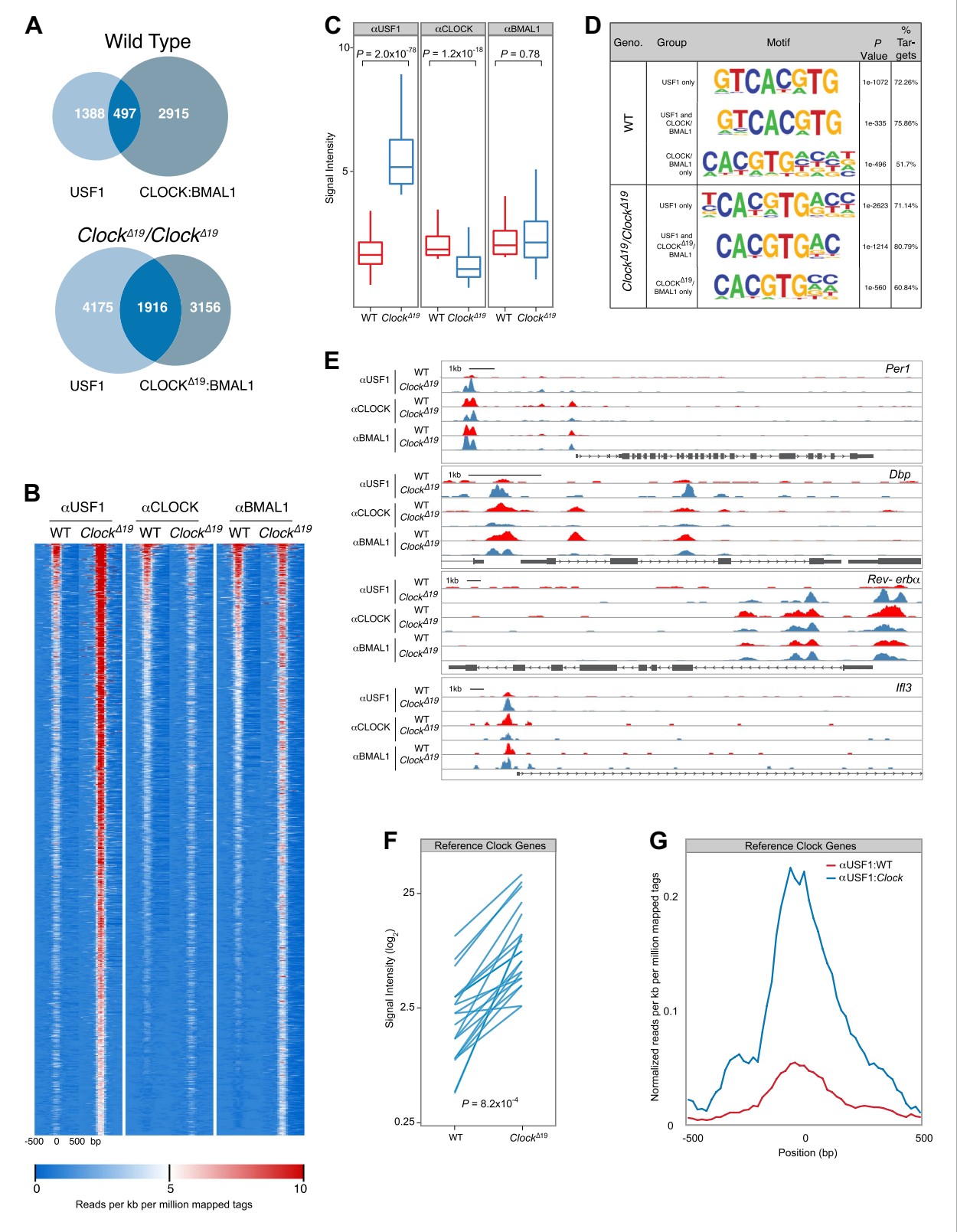

**Figure 8**. Genome-wide location analysis (Chip-Seq) reveals extensive overlap between USF1 and CLOCK or BMAL1 binding sites. (**A**) Venn diagram showing a dramatic increase in sites bound by both USF1 and CLOCK:BMAL1 in $Clock^{\Delta 19}$ mutant animals (bottom) as compared to wild-type animals (top). (**B**) Heat map analysis of the 1916 common sites between USF1 and CLOCK:BMAL1 reveals an increase in the number of USF1 binding sites
*Figure 8. Continued on next page*

*Figure 8. Continued*

as well as binding intensity at existing sites in *Clock^Δ19* mutant animals (left two heat maps). CLOCK binding decreases in *Clock^Δ19* mutant animals (middle two heat maps), while BMAL1 binding is largely unchanged (right two heat maps). (**C**) Quantification of the top 200 binding sites for each transcription factor is represented as box plots. Similar to the heat map analysis, USF1 binding increases (left panel), CLOCK binding decreases (middle panel), and BMAL1 binding is unaffected in *Clock^Δ19* mutant mice (right panel). (**D**) Motif analysis of each of the six groups of binding sites reveals a common canonical E-box CACGTG sequence. (**E**) UCSC browser view of binding peaks at four representative genes *Per1*, *Dbp*, *Rev-erbα*, and *Ilf3*. Each horizontal track represents the ChIP-seq binding signal as described previously (*Koike et al., 2012*) for either WT (red) or *Clock^Δ19* mutants (blue) for USF1, CLOCK or BMAL1 as indicated on the left. In each case, USF1 binding increases in *Clock^Δ19* mutants. (**F**) Binding signal of USF1 increases at most reference clock genes in *Clock^Δ19* mutant animals. (**G**) Averaged histogram view of reference clock genes 1 kb around peak of binding also shows increased binding in *Clock^Δ19* mutant mice. The ChIP-seq peak lists for USF1, CLOCK and BMAL1 are available in *Figure 8—source data 1*, *2* and *3* respectively.

The following source data are available for figure 8:

**Source data 1.** ChIP-seq peak list for USF1.
**Source data 2.** ChIP-seq peak list for CLOCK.
**Source data 3.** ChIP-seq peak list for BMAL1.

## Bioluminescence recording

*Per2^Luc* mice (*Yoo et al., 2004*) were euthanized by cervical dislocation between ZT11 and ZT12.5. Tissue was rapidly dissected as described in (*Yamazaki and Takahashi, 2005*). 300 µm coronal sections containing the SCN were isolated. Tissues were cultured on Millicell culture membranes (PICMORG50, Millipore, Billerica, MA) and were placed in 35 mm tissue culture dishes containing 2 ml DMEM media (90-013-PB, Gibco, Life Technologies, Grand Island, NY) supplemented with 352.5 µg/ml sodium bicarbonate, 10 mM HEPES (Gibco), 2 mM L-Glutamine, 2% B-27 Serum-free supplement (Invitrogen, Life Technologies, Grand Island, NY), 25 units/ml penicillin, 25 µg/ml streptomycin (Gibco), and 0.1 mM luciferin potassium salt (L-8240, Biosynth AG, Staad, Switzerland). Sealed cultures were placed in LumiCycle luminometer machines (Actimetrics, Wilmette, IL) and bioluminescence from the tissue was recorded continuously.

## Clock suppressor mapping cross

(BALB/cJ x C57BL/6J)F2 *Clock^Δ19*/+ mice were bred by intercrossing (BALB/cJ x C57BL/6J)F1 *Clock^Δ19*/+ animals. Only *Clock^Δ19*/+ mice were wheel-tested. Note that because *Clock* maps to chromosome 5, and the mutation was originally induced in the B6 background, it was not possible to scan for QTLs on this chromosome in the F2 cross. Genotyping primers used were: forward 5′-TACCAGCTGCTAATGTCCAGTG-3′; reverse 5′-TACATTGGGCTAGCCTTCCTAAG-3′. PCR conditions were 95°C for 2 min followed by 32 cycles of 95°C for 15 s, 55°C for 30 s, and 72°C for 15 s. Following amplification, PCR products were digested with 2 units of HincII for 2 hr at 37°C. After restriction digestion, PCR products were resolved on 4% agarose gels. The *Clock* mutant allele was identified as a ~70-bp product, while wild-type animals were identified by the presence of bands at ~50 and 20 bp.

## QTL analysis

SSLP markers used in this study are available upon request. The genotyping protocol has been described previously (*Shimomura et al., 2001*). QTL analysis was performed using MapmanagerQTX (http://www.mapmanager.org/).

## Congenic lines

(BALB/cJ x C57BL/6J)F1 *Clock^Δ19*/+ mice were backcrossed to C57BL/6J wild-type mice. Animals were genotyped with five SSLP markers (*D1Mit22*, *D1Mit218*, *D1Mit33*, *D1Mit291* and *D1Mit155*) to identify individuals carrying BALB alleles of *Soc*. Following the N4 generation, we selected mice for breeding by both phenotype and genotype. After the N6 generation, selection was performed by genotyping with *D1Mit452* and *D1Mit155*. Mice carrying smaller fragments of the *Soc^BALB* region and with suppressed phenotypes were crossed to C57BL/6J wild-type animals. After the 10th generation of phenotypic and genotypic selection, mice were intercrossed to produce the final congenic line.

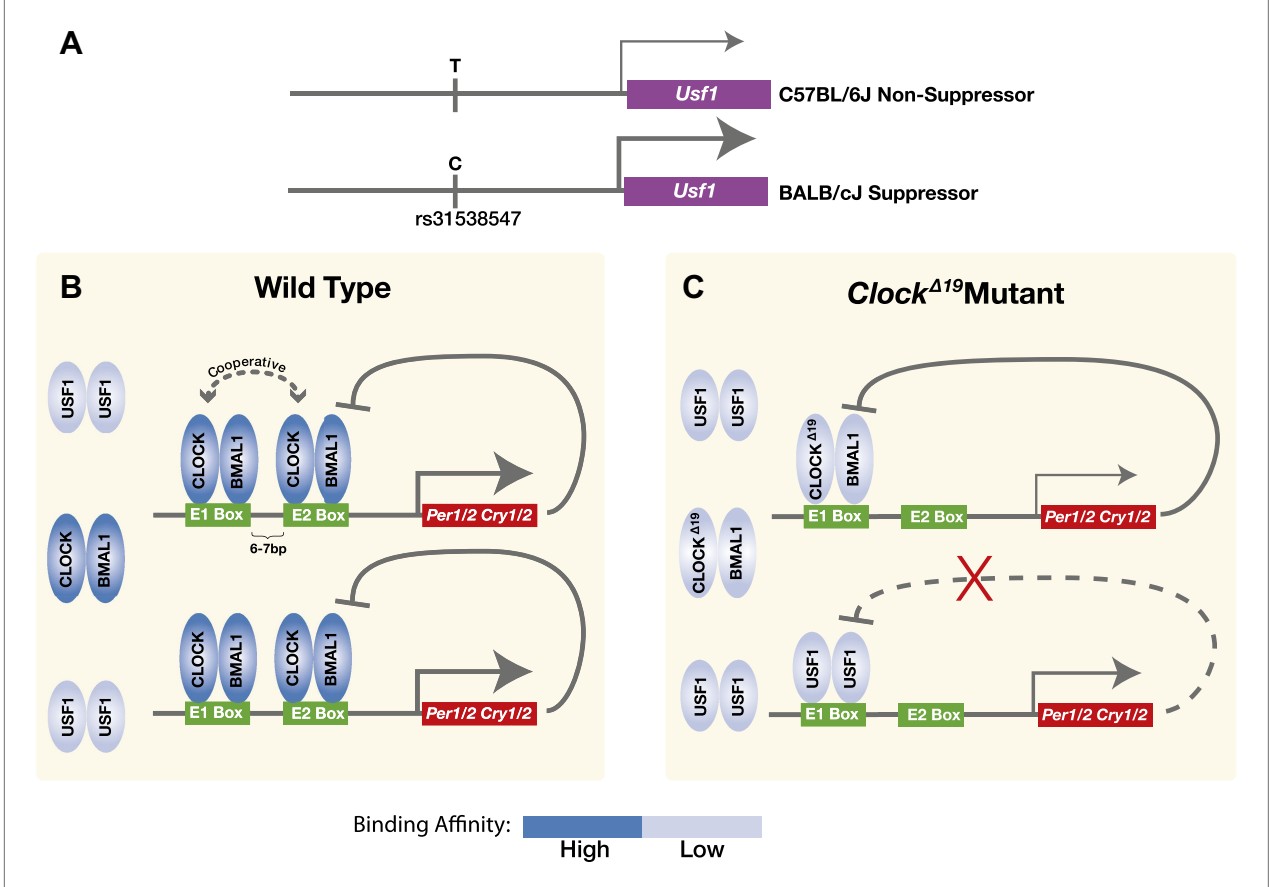

**Figure 9**. Model for the USF1 suppression of the *Clock^Δ19* mutation within the circadian clock mechanism. (**A**) SNP polymorphisms between B6 and BALB in the *Usf1* promoter lead to a *cis*-mediated increase in *Usf1* expression levels in suppressor mouse strains. (**B**) In the wild-type state, CLOCK:BMAL1 tandem heterodimers bind to tandem E-box sites in the regulatory regions of the *Per* and *Cry* genes and interact cooperatively to enhance binding affinity which is much higher than USF1 affinity. Thus, most tandem E-box sites are preferentially bound by wild-type CLOCK:BMAL1 complex over USF1. (**C**) In contrast, in the *Clock^Δ19* mutant, the CLOCK^Δ19:BMAL1 complex binds to the E1 site primarily as a single heterodimer with an affinity similar to that of USF1, yet lower than that of the wild-type CLOCK:BMAL1 tandem heterodimer complex. This allows USF1 to occupy CLOCK:BMAL1 sites more effectively in the mutant state and, as a consequence, drive transcription of CLOCK:BMAL1 target genes under conditions where mutant CLOCK^Δ19:BMAL1 is not transcriptionally competent. Because USF1 is not repressed by CRY1 or CRY2 negative feedback, the transcriptional activation by USF1 may be more responsive than that of CLOCK:BMAL1. However, negative feedback can still occur via CLOCK^Δ19:BMAL1 since circadian oscillations can persist in *Clock^Δ19* mutants.

### Interval-specific haplotype association

All inbred strains were purchased from Jackson Laboratory (Bar Harbor, ME). These strains were crossed to C57BL/6J *Clock^Δ19*/+ females. For high-resolution mapping of the *Soc* locus, we used the 150 K SNP set from The Genomics Institute of the Novartis Foundation.

### Quantitative PCR (qPCR)

RNA was extracted using Trizol reagent (Invitrogen) according to the manufacture's protocol. RNA was then reverse-transcribed into cDNA by TaqMan RT kit. qPCRs were performed with SYBR green in an Applied Biosystems 7900HT Fast Real-Time PCR System (384 well). Primer sequences are described in *supplementary file 1*. Copy number estimates were performed as described previously (*Uno and Ueda, 2007*).

### Allele-specific expression

We used three different methods for detecting B6 and BALB allele specific expression in (BALB x B6) F1 *Clock^Δ19*/+ animals.

## Sequencing method

We amplified the region surrounding the SNP at exon 10 (rs31093636). The amplicon sequence between genomic DNA and cDNA is identical. We PCR-amplified the SNP-containing region in F1 cDNA and genomic DNA with several different mix ratios between B6 and BALB. The PCR products were sequenced on an Applied Biosystems 3130xl Genetic Analyzer. We used the fluorescent signal peak ratio as an indicator of the copy number ratio of the two alleles. To minimize genetic background effects, we used heterozygous congenic mouse cDNA. These mice are C57BL/6J isogenic except for ~25 Mb (165.1–190.45 Mb) of chromosome 1.

## qPCR method

We also used allele discrimination with allele-specific primers at SNP on exon 8 (rs31541670). We initially tested BALB and B6 genomic DNA mixtures to create a standard curve (%BALB allele vs $\Delta C_t$ between BALB and B6 allele-specific primers). Then we amplified cDNA from F1 hypothalamus RNA to measure $\Delta C_t$. By using the previously made standard curve, %BALB allele was determined.

## Colony counting method

F1 genomic DNA and F1 cDNA were amplified by using primers flanking the SNP at exon 10 (rs31093636). PCR products were cloned into a TOPO TA cloning vector (Invitrogen). This SNP creates a restriction fragment length polymorphism. The BALB, but not the B6 allele, is recognized by TfiI. We had four independent PCR products from single F1 genomic DNA and four PCR products from four different F1 cDNA samples. Each PCR product was cloned into the TOPO TA cloning vector and transformed into DH5α *E. coli*. From each transformation, we picked 24–48 colonies to determine the percentage of colonies containing the BALB allele. Primer sequences are described in *supplementary file 1*.

## Transactivation assays

HEK293T cells were cultured at 37°C under a humidified atmosphere containing 5% $CO_2$ in Dulbecco's modified Eagle's medium supplemented with 10% fetal bovine serum. We seeded cells into 24-well poly-D-lysine coated culture plates (~2 × 10$^5$ cells/well). After 24 hr, plasmid DNAs were transiently transfected into the HEK293T cells using Lipofectamine2000 reagent (Invitrogen) according to the manufacturer's protocol. After 24 hr of transfection, the cells were harvested and luciferase assay was performed and the raw numbers were normalized with a beta-actin-LacZ transfection control.

## Usf1 transgenic mice

Mouse *Usf1* cDNA clone (MGC:59,374) was obtained from the Mammalian Gene Collection (Open Biosystems) and was used to PCR-amplify *Usf1* with forward (5′-GCCGCCACCATGAAGGGGC-AGCAGAAAA-3′) and reverse primers (5′-TTAAAGAGCGTAATCTGGAACATCGTATGGGTAGTT-GCTGTCATTCTTGATGACGACCT-3′). The PCR product containing a modified *Usf1* with a Kozak consensus sequence and a 3′ HA tag was inserted into pBluescript II KS- using standard cloning procedures. Site-directed mutagenesis was performed with forward (5′- GCAGGGAGGGAGCCAGCGATC-3′) and reverse (5′- GATCGCTGGCTCCCTCCCTGC -3′) primers using Quikchange XL Site-Directed Mutagenesis Kit (Stratagene, La Jolla, CA). The product was reamplified with forward (5′-GCCGCCACC-ATGAAGGGGCAGCAGAAAA-3′) and reverse (5′-TTAAAGAGCGTAATCTGGAACATCGTATGGGT-AGTTGCTGTCATTCTTGATGACGACCT-3′) primers and subcloned into the EcoRV restriction site in PMM-400 (provided by Mark Mayford). Transgenic mouse lines were generated by pronuclear injection using standard techniques as described (*Hong et al., 2007*). The linearized DNA fragment was injected at a concentration of 1 ng/µl into fertilized mouse oocytes isolated from crosses of C57BL/6J matings to produce transgenic mice that were isogenic on C57BL/6J to exclude the possibility of contamination by genetic suppressor backgrounds. Transgenic mice were identified by PCR analysis of genomic DNA prepared from tail biopsy samples. Two sets of primers were used for genotyping: USF1-5F: 5′-CCAAAAACGAGGAGGATTTG-3′ and USF1-5R: 5′-GTGGCAGGGTAACCACTGAT-3′, or USF1-3F: 5′-GCAGGGGTTAGATCAGTTGC-3′ and USF1-3R: 5′-TGCTCCCATTCATCAGTTCC-3′. PCR conditions were: 95°C for 2 min followed by 32 cycles of 95°C for 15 s, 55°C for 30 s, and 72°C for 30 s. Following amplification, PCR products were resolved on 2% agarose gels.

## Chromatin immunoprecipitation (ChIP)

Livers from mice were immediately homogenized in 4 ml per liver of 1× PBS containing 2 mM EGS. The homogenate was incubated for 20 min at room temperature. Next, 110 µl of 36.5% formaldehyde was

added followed by an 8 min incubation at room temperature. Cross-linking reactions were stopped by the addition of 250 µl of 2.5 M glycine and kept on ice. The homogenate was added to 10 ml of ice-cold 2.3 M sucrose containing 150 mM glycine, 10 mM HEPES pH 7.6, 15 mM KCl, 2 mM EDTA, 0.15 mM spermine, 0.5 mM spermidine, 0.5 mM DTT and 0.5 mM PMSF, and layered on top of a 3 ml cushion of 1.85 M sucrose (containing the same ingredients with 10% glycerol) and centrifuged for 1 hr at 24,000 rpm at 4°C in a Beckman SW32.1 rotor. The nuclei were resuspended in 1 ml of 20 mM Tris pH 7.5, 150 mM NaCl, 2 mM EDTA, transferred to a 1.5 ml microfuge tube, centrifuged for 3 min at 3000 rpm at 4°C, washed again, and stored –80°C until use. The nuclei were then resuspended in 0.8 ml per liver of lysis buffer (10 mM Tris pH 7.5, 1 mM EDTA, 0.5 mM EGTA, 0.2 M NaCl, 0.5% *N*-lauroylsarcosine, 0.1% sodium deoxycholate, 1 mM PMSF and EDTA-free protease inhibitor cock-tail; Roche, Indianapolis, IN), and sonicated 5 s at 4°C 36 times with a Covaris S2. The fragmented chromatin was then diluted tenfold with IP buffer (10 mM Tris pH 7.5, 150 mM NaCl, 1 mM EDTA, 1% triton X-100, 0.1% sodium deoxycholate, 1 mM PMSF, EDTA-free protease inhibitor cocktail). Approximately 120 µg of fragmented chromatin was pre-cleared by incubating with 40 µl of protein A-agarose (Sigma, St. Louis, MO) for 2 hr at 4°C with rotation. Pre-cleared chromatin was then incubated with antibody overnight at 4°C with rotation, followed by addition of 10 µl of Protein A/G Plus-agarose and incubation for 1.5 hr at 4°C. Agarose beads were then washed twice with IP buffer, twice with high salt wash buffer (20 mM Tris pH7.5, 500 mM NaCl, 2 mM EDTA, 1% Triton X-100, 1 mM PMSF), twice with LiCl wash buffer (20 mM Tris pH 7.5, 250 mM LiCl, 2 mM EDTA, 0.5% Igepal CA-630, 1% sodium deoxycholate, 1 mM PMSF), and once with TE. Co-immunoprecipitated DNA fragments were eluted with 100 µl of 20 mM Tris pH 7.5, 5 mM EDTA, 0.5% SDS, then reverse crosslinked at 65°C overnight, incubated with 10 µg of RNaseA for 30 min at 37°C, with 160 µg of proteinase K for 30 min at 55°C, and then purified using the QIAquick PCR purification Kit (Qiagen, Germantown, MD). ChIP products were analyzed by qPCR using SYBR green. Primer sequences are described in *supplementary file 1*.

## Chromatin immunoprecipitation sequencing (ChIP-Seq)

ChIP-Seq libraries were prepared as described (*Kim et al., 2010*; *Koike et al., 2012*). SOLiD sequencing of ChIP-Seq libraries were performed on an ABI SOLiD4 instrument with 35-bp reads according to manufacturer's instructions (Life Technologies, Grand Island, NY) by the UTSW McDermott DNA Sequencing Core. Sequence reads were mapped to the mouse genome (NCBI m37) with Applied Biosystems BioScope v1.3. The peaks were identified from uniquely mapped reads without duplicates using MACS (*Zhang et al., 2008*) followed by PeakSplitter (*Salmon-Divon et al., 2010*). Sequence reads were checked for abnormal read content in input samples that led to global false positive peak detection that arise primarily from low complexity sequence and contamination from plasmid/cDNA sequences. Global false positive peaks were removed from peak lists derived from MACS and PeakSplitter, and tags derived from plasmids/cDNAs were not analyzed as described previously (*Heinz et al., 2010*). Results were further analyzed using HOMER (*Heinz et al., 2010*). Peak overlaps (peak summit ± 120 bp) were determined with HOMER. Genome browser views were normalized as uniquely mapped reads per 10 million reads.

## Preparation of mouse liver nuclear proteins

Liver nuclear extract was prepared as described previously (*Yoshitane et al., 2009*). Mice were sacrificed by cervical dislocation at ZT8. Whole liver was quickly removed and briefly washed with ice-cold PBS and homogenized with seven strokes of a dounce tissue homogenizer (Pestle A only) at 4°C in 9 ml of ice-cold buffer A (10 mM HEPES-NaOH [pH 7.8], 10 mM KCl, 0.1 mM EDTA, 1 mM dithiothreitol [DTT], 1 mM phenylmethylsulfonyl fluoride [PMSF], 4 mg/ml aprotinin, 4 mg/ml leupeptin, 50 mM NaF, and 1 mM Na$_3$VO$_4$). The homogenate was centrifuged twice (5 min each, 700×$g$), and the precipitate was resuspended in 2 ml of ice-cold buffer C (20 mM HEPES-NaOH [pH 7.8], 400 mM NaCl, 1 mM EDTA, 5 mM MgCl$_2$, 2% [v/v] glycerol, 1 mM DTT, 1 mM PMSF, 4 mg/ml aprotinin, 4 mg/ml leupeptin, 50 mM NaF, and 1 mM Na$_3$VO$_4$). After gentle mixing at 4°C for 30 min, the suspension was centrifuged twice (30 min each at 21,600×$g$), and the final supernatant was used as a nuclear extract.

## Electrophoretic mobility shift assay (EMSA)

EMSA was performed in a manner similar to that described previously (*Ripperger and Schibler, 2006*). For oligonucleotide annealing, 1 nm of each forward and reverse oligonucleotide (*supplementary file 1*) were mixed with 1× T4 polynucleotide buffer at final volume 50 µl. The reaction mix was heated at 95°C for 10 min then slowly cooled to 25°C (1 °C/min). Oligonucleotide labeling was performed in

50 µl with 50 pmole of double-stranded oligo, 5 µl of T4 polynucleotide kinase buffer (NEB, Ipswich, MA), 2.5 µl of T4 polynucleotide kinase (NEB), 3.75 µl ATPγ$^{32}$P (Perkin Elmer cat# BLU002Z250UC, Waltham, MA), and 36.25 µl water. The reaction mix was incubated at 37°C for 30 min followed by 65°C for 10 min. After the kinase reaction, the sample was purified through a G50 spin column. EMSA reactions were performed with 15 µg of nuclear protein in 25 mM HEPES-KOH (pH 7.6), 150 mM NaCl, 0.1 mM EDTA, 1 mM DTT, 200 ng/µl sheared salmon sperm DNA, 50 ng/µl poly(dI-dC), and 2 µl of radioactive probe in a volume of 12 µl. Binding buffer and nuclear protein were mixed and incubated at room temperature for 15 min. Then, 2 µl of radioactive probe was added and incubated at 16°C for 30 min. For supershifts, 1 µl of antibody was added immediately before the addition of the radio-labeled probe. The antibodies used were anti-BMAL1, anti-CLOCK (a gift from Dr. Choogon Lee) as described previously (*Koike et al., 2012*), and anti-USF1 (sc-8983×, Santa Cruz Biotech, Dallas, TX). 1 µl of 15% Ficoll was added, and the reaction mixes were loaded on 4% polyacrylamide gels (0.5× TBE with 2.5% glycerol). Gels (0.2 mm thickness, 4% polyacrylamide, Bio-Rad #161–0144) were run for 90 min at room temperature (7.5 V/cm) and exposed on phosphor imager screens for 18 hr at 4°C. For saturation binding assays, increasing amounts of probe were added to a fixed amount of nuclear extract in triplicate. A four-parameter nonlinear least squares curve fitting method (Prizm, Graphpad Software, La Jolla, CA) was used to estimate the $B_{max}$ and $K_d$ values and to conduct tests for whether each best-fit parameter differs among the four groups in *Table 2*.

## Usf1 gene trap mouse production

Mouse embryonic stem cell gene trap line W233A07 and P116C05 were obtained from the German Gene Trap Consortium (http://www.genetrap.de). The ROSAbetageo+1 gene trap vector (incorporating fused beta galactosidase and neomycin) was introduced to 129S2/SvPas-derived TBV2 embryonic stem (ES) cells to generate mutant cell lines. Both W233A07 and P116C05 cells, carrying a vector insertion into the intron one of *Usf1* gene, were used to generate mice which were backcrossed to C57BL/6J for four generations. In order to detect presence of trap vector, *Usf1*-TrapFwd: 5'-AGTGACAACGTCGAGCACAG-3', *Usf1*-TrapRev: 5'-CGGTCGCTACCATTACCAGT-3' were used. To discriminate +/− from −/− mouse, we used real time PCR using *Usf1*-Trap 129Fwd:5'- GGAGGT-GGGGATTATAGTCTGAA-3', *Usf1*-Trap B6Fwd:5'- GGAGGTGGGGATTATAGTCTGAG-3' and *Usf1*-Trap Rev:5'-TGCTACAAGGAGGGGTTCTG-3'. Since mutant allele at *Usf1* locus is derived from 129S2/SvPas, +/− mouse is 129/B6 heterozygous and +/− mouse is 129/129 homozygous at *Usf1* locus.

## Acknowledgements

We thank Ueli Schibler and Jurgen Ripperger for advice and discussion, Helen Hobbs, Vanessa Schmidt and William Crider of the UT Southwestern McDermott DNA Sequencing Core for next generation sequencing support, Fred W Turek for support, Hung-Chung Huang for bioinformatics support, Peter D Zemenides and Andrew Lin for genotyping, and Choogon Lee for CLOCK antibody. KS, VK and SHY were Howard Hughes Medical Institute Research Associates. JST is a Howard Hughes Medical Institute investigator.

## Additional information

### Funding

| Funder | Grant reference number | Author |
|---|---|---|
| Howard Hughes Medical Institute | H012233 | Joseph S Takahashi |
| National Institutes of Health | R37 MH39592, U01 MH61915, P50 MH074924 | Joseph S Takahashi |
| National Science Foundation Science and Technology Center for Biological Timing | DIR-8920162 | Joseph S Takahashi |
| National Institutes of Health | F32 DA024556 | Vivek Kumar |
| National Institutes of Health | R15 GM086825 | Phillip L Lowrey |

| Funder | Grant reference number | Author |
|---|---|---|
| Novartis Research Foundation | | Mathew T Pletcher |
| Novartis Research Foundation | | Tim Wiltshire |
| Novartis Research Foundation | | John Hogenesch |
| Whitehall Foundation | | Tae-Kyung Kim |
| Chicago Biomedical Consortium | | Joseph Bass |
| University of Chicago Diabetes Research and Training Center | DK-20595 | Joseph Bass |

The funders had no role in study design, data collection and interpretation, or the decision to submit the work for publication.

### Author contributions

KS, Conception and design, Acquisition of data, Analysis and interpretation of data, Drafting or revising the article; VK, Acquisition of data, Analysis and interpretation of data, Drafting or revising the article; NK, EDB, MTP, TW, JC, ARW, SSL, CO, DF, JRO, MR, S-HY, H-KH, MHV, Acquisition of data, Analysis and interpretation of data; T-KK, JB, JH, Conception and design, Analysis and interpretation of data; PLL, Analysis and interpretation of data, Drafting or revising the article; JST, Conception and design, Analysis and interpretation of data, Drafting or revising the article

### Ethics

Animal experimentation: All animal care and use procedures were in accordance with guidelines of the Northwestern University and UT Southwestern Institutional Animal Care and Use Committees.

## Additional files

### Supplementary files

• Supplementary file 1. Oligonucleotide sequences.

### Major datasets

The following dataset was generated:

| Author(s) | Year | Dataset title | Dataset ID and/or URL | Database, license, and accessibility information |
|---|---|---|---|---|
| Shimomura K, Koike N, Takahashi JS | 2012 | *Usf1*, a suppressor of the circadian *clock* mutant, reveals the nature of the DNA-binding of the CLOCK:BMAL1 complex | GSE44609; http://www.ncbi.nlm.nih.gov/geo/query/acc.cgi?acc=GSE44609 | Publicly available at the NCBI Gene Expression Omnibus (http://www.ncbi.nlm.nih.gov/geo/) |

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
