## [Decision Letter]

Thank you for choosing to send your work entitled “*Usf1*, a Suppressor of the Circadian *Clock* Mutant, Reveals the Nature of the DNA-binding of the CLOCK:BMAL1 complex” for consideration at *eLife*. Your article has been favorably evaluated by a Senior editor and 3 reviewers, one of whom is a member of our Board of Reviewing Editors.

The following individuals responsible for the peer review of your submission want to reveal their identity: Louis Ptáček, Reviewing editor; Ueli Schibler, peer reviewer.

The reviewers very much appreciated the work presented and the Reviewing editor has assembled the following comments to help you prepare a revised submission.

Using a heroic forward genetics approach, Takahashi's group has previously identified a semidominant mutation in the *Clock* gene that lengthens the circadian period in heterozygous and homozygous mice of the C57BL/6J (B6) strain (King et al., 1997). Moreover, when exposed to constant darkness over an extended time period, most homozygous *Clock* mutant mice become arrhythmic. The CLOCK mutant protein still contains functional dimerization and DNA binding domains, but fails to efficiently activate target gene transcription.

In the present study, Shimomura and coworkers extended this work by identifying a non-allelic suppressor (*Soc*) of the *Clock* mutant allele as *Usf1* (*upstream stimulatory factor 1*) in BALB and other mouse strains. To this end they used a clever combination of QTL and haplotype block mapping to narrow the chromatin region carrying the *Soc* locus down to a DNA segment encompassing about 900 Kb and 22 protein encoding genes. A transcript expression analysis for seven candidate genes with a similar spatial expression pattern as *Clock* revealed that only one of these, *Usf1*, displayed significantly higher transcript accumulation in mice with the *Soc* phenotype. Moreover, a moderate overexpression of USF1 from a transgene driven by the minimal CMV promoter in B6 mice suppressed the *Clock* mutant phenotype in a similar fashion as the BALB *Usf1* allele. The cis-acting region responsible for increased *Usf1* transcription in BALB mice has been mapped to an about 1kb DNA fragment in the 5'-flanking region of *Usf1*. Importantly, in vitro (EMSA) and in vivo (ChIP-seq) binding studies demonstrated that USF1 and CLOCK-BMAL1 share significant DNA-binding specificity. The actograms of voluntary locomotor activity of *Usf1* knockout mice do not show a significantly different period length, but display lower amplitude and overall activity when compared to those of wild-type mice. This suggests that USF1 also contributes to the robustness of circadian behavior in the latter.

1. The authors argue USF1 binds to tandem E-box and maintains the open chromatin structure when CLOCK/BMAL1 heterodimer levels decline at night. This is an ideal conclusion of the manuscript, but no direct proof is offered. DNAse hypersensitivity or equivalent technique in WT, *Clock* mutant or *Usf1* knockout mice will strengthen the conclusion. If the authors do not have this expertise, they may simply remove this claim. If they want to pursue this conclusion, they may also test the following two scenarios.

2. The above role would suggest USF1 functions in a cell autonomous manner and its overexpression or knockdown should produce period/amplitude alteration of the clock. The authors argue the reduced activity and low amplitude activity rhythm supports such function. However, there are potentially alternate modes of action that can produce such a phenotype in a whole animal. For example, *Usf1* knockout might reduce expression of a synchronizing signal in the SCN leading to reduced amplitude of the behavioral rhythms. Overexpression or knockdown experiments in cell lines or single cell monitoring of bioluminescence from *Usf1*KO;*Per2*^*Luc*^ mice will be better to test this hypothesis.

3. If the model is true, *Usf1*KO in *Clock* mutant background would produce a more severe phenotype than the mere additive effects of two mutations. If the phenotype is comparable to that of *Clock* mutant, it would suggest either the proposed role of USF1 in maintaining open chromatin structure is not essential or another USF family member can compensate for the loss of USF1. Testing this double mutant is therefore critical in resolving the role of USF family of proteins in the circadian system.

4. The ChIP experiments are done in *Clock* homozygous mice while the *Clock* heterozygote mice were used to test for suppressor effect of Balbc *Usf1* allele. ChIp results in heterozygote may be added.

---

## [Author Response]

*1. The authors argue USF1 binds to tandem E-box and maintains the open chromatin structure when CLOCK/BMAL1 heterodimer levels decline at night. This is an ideal conclusion of the manuscript, but no direct proof is offered. DNAse hypersensitivity or equivalent technique in WT,* Clock *mutant or* Usf1 *knockout mice will strengthen the conclusion. If the authors do not have this expertise, they may simply remove this claim. If they want to pursue this conclusion, they may also test the following two scenarios*.

Since the statement “The role of USF1 at CLOCK:BMAL1 sites may therefore be to maintain an open chromatin state to facilitate CLOCK:BMAL1 binding on the following circadian cycle” was made in the Discussion section, we did not intend the statement to be a conclusion, but rather speculation. In order to make our speculation crystal clear, we have modified the statement to read “We speculate that the role of USF1 at CLOCK:BMAL1 sites could be to maintain an open chromatin state to facilitate CLOCK:BMAL1 binding on the following circadian cycle although additional work would be required to test this hypothesis.”

Since it seems reasonable to be able to put forward some ideas in the Discussion section, we would like to be able use this qualified language in the revision.

*2. The above role would suggest USF1 functions in a cell autonomous manner and its overexpression or knockdown should produce period/amplitude alteration of the clock. The authors argue the reduced activity and low amplitude activity rhythm supports such function. However, there are potentially alternate modes of action that can produce such a phenotype in a whole animal. For example,* Usf1 *knockout might reduce expression of a synchronizing signal in the SCN leading to reduced amplitude of the behavioral rhythms. Overexpression or knockdown experiments in cell lines or single cell monitoring of bioluminescence from* Usf1*KO;*Per2^Luc^
*mice will be better to test this hypothesis*.

We agree that the *Usf1* knockout mouse circadian activity experiments cannot be interpreted vis-à-vis the underlying mechanism. Thus, we have inserted the following statement to make this point explicitly and so as not to imply that we know how *Usf1* loss-of-function affects the cell autonomous circadian clock:

“While it is tempting to speculate that these effects of *Usf1* might work at the level of the cell autonomous circadian oscillator, additional experiments would be required to determine at what level of organization (cell, circuit or higher) these changes in amplitude at the behavioral level originate.”

Since this point refers back to the speculation in the previous point, and we have qualified the statements in both, we ask that the editors to allow some leeway in discussing these issues in the revision.

*3. If the model is true,* Usf1*KO in* Clock *mutant background would produce a more severe phenotype than the mere additive effects of two mutations. If the phenotype is comparable to that of* Clock *mutant, it would suggest either the proposed role of USF1 in maintaining open chromatin structure is not essential or another USF family member can compensate for the loss of USF1. Testing this double mutant is therefore critical in resolving the role of USF family of proteins in the circadian system*.

Again this point refers back to the speculation in the first point, which we have qualified, and therefore, we would assert that it is not our intention in this paper to analyze definitively or to resolve the “*role of USF family of proteins in the circadian system”* independently from the work already presented here showing that USF1 is a suppressor of the *Clock* mutant. We agree with the suggestion that these experiments would be interesting to undertake to explore the interactions of USF1 and CLOCK in the future.

However, we hope the editors agree that the double mutant experiments go beyond the scope of this already rather extensive manuscript. Such double mutant experiments would be extremely onerous at this time because we have frozen down the *Usf1* knockout mouse line, and this would require new matings to obtain double homozygous, double mutant mice (which are very inefficient crosses using the *Clock* mutant, which does not breed as a homozygote and is compromised even as a heterozygote: we have previously documented the reproductive defect in female *Clock* mutant mice in Miller et al. 2004. Circadian *Clock* mutation disrupts estrous cyclicity and maintenance of pregnancy. *Current Biology* 14:1367-1373.) Such double homozygous, double mutant mice would take us 6–12 months to produce, adding a very significant delay to this work.

*4. The ChIP experiments are done in* Clock *homozygous mice while the* Clock *heterozygote mice were used to test for suppressor effect of Balbc* Usf1 *allele. ChIp results in heterozygote may be added*.

We agree that it would be desirable for us to have ChIP data from heterozygous *Clock*^Δ19^*/+* mice, but we did not attempt this work since we did not know whether the biochemical assays would be sensitive enough to detect differences in heterozygous mice given the subtle nature of the genetic suppression. To illustrate the subtlety, the BALB suppression of *Clock*^Δ19^*/+* period length is ∼0.3 hours, which is detectable at the behavioral level because the precision of the circadian clock is so good. However, this difference in period corresponds to a 0.3/24 = 1.25% difference in the cycle length. At the molecular level, we cannot detect such subtle changes and even if the molecular changes were 10-fold greater, they would be hard to discern. We also did not attempt biochemical experiments on heterozygous *Clock*^Δ19^*/+* mice because we cannot discriminate between wild-type CLOCK and CLOCK^Δ19^ in ChIP experiments and because the EMSA mobility patterns were too complex to interpret with both CLOCK:BMAL1 and CLOCK^Δ19^:BMAL1 complexes in the same lanes.

Also, we do not make any assertions in the manuscript that imply that the experiments might be construed as having been performed on heterozygous mice. The ChIP and EMSA experiments were intended to explore the effects of the CLOCK^Δ19^ mutation on the binding affinity of the CLOCK:BMAL1 complex and to determine whether CLOCK^Δ19^ affects USF1 DNA occupancy. We view these experiments as intended to explore the biochemical mechanism of CLOCK:BMAL1 and CLOCK^Δ19^:BMAL1 occupancy in a manner analogous to the use of in vitro biochemical assays to explore mechanism in a more simplified state, rather than to recapitulate the complex *Clock*^Δ19^*/+* heterozygous state. Therefore, we would ask the editors to consider the addition of ChIP experiments on heterozygous mice as beyond the scope of this work and too onerous to undertake at this time.